# Mismatched No More:
# Joint Model-Policy Optimization for Model-Based RL

**Benjamin Eysenbach**[* 1 2]    **Alexander Khazatsky**[* 3]    **Sergey Levine**[2 3]    **Ruslan Salakhutdinov**[1]

[1]Carnegie Mellon University,    [2]Google Brain,    [3]UC Berkeley
beysenba@cs.cmu.edu,   khazatsky@cs.stanford.edu

## Abstract

Many model-based reinforcement learning (RL) methods follow a similar template: fit a model to previously observed data, and then use data from that model for RL or planning. However, models that achieve better training performance (e.g., lower MSE) are not necessarily better for control: an RL agent may seek out the small fraction of states where an accurate model makes mistakes, or it might act in ways that do not expose the errors of an inaccurate model. As noted in prior work, there is an objective mismatch: models are useful if they yield good policies, but they are trained to maximize their accuracy, rather than the performance of the policies that result from them. In this work, we propose a single objective for jointly training the model and the policy, such that updates to either component increase a lower bound on expected return. To the best of our knowledge, this is the first lower bound for model-based RL that holds globally and can be efficiently estimated in continuous settings; it is the only lower bound that mends the objective mismatch problem. A version of this bound becomes tight under certain assumptions. Optimizing this bound resembles a GAN: a classifier distinguishes between real and fake transitions, the model is updated to produce transitions that look realistic, and the policy is updated to avoid states where the model predictions are unrealistic. Numerical simulations demonstrate that optimizing this bound yields reward maximizing policies and yields dynamics that (perhaps surprisingly) can aid in exploration. We also show that a deep RL algorithm loosely based on our lower bound can achieve performance competitive with prior model-based methods, and better performance on certain hard exploration tasks.

## 1   Introduction

Much of the appeal of model-based RL is that model learning is a simple and scalable supervised learning problem. However, even after learning a very accurate model, it is hard to say whether that model will actually be useful for model-based RL [17, 29]. For example, a model might make small mistakes in critical states that cause a policy to take suboptimal actions. Alternatively, a model with large errors may yield a policy that attains high return if the model errors occur in states that the policy never visits.

The underlying problem is that dynamics models are trained differently from how they are used. Typical model-based methods train a model using data sampled from the *real* dynamics (e.g., using maximum likelihood), but apply these models by using data sampled from the *learned* dynamics [11, 23, 24, 48]. Prior work has identified this *objective mismatch* issue [17, 29, 30]: the model is trained using one objective, but the policy is trained using a different objective. Designing an objective for model training that is guaranteed to improve the expected reward remains an open problem.

---

[*]Equal contribution.

36th Conference on Neural Information Processing Systems (NeurIPS 2022).

So, *how should we train a dynamics model so that it produces high-return policies when used for model-based RL?*

The key idea in this paper is to view model-based RL as a latent-variable problem: the latent variable is the trajectory and the cumulative reward is interpreted as the probability that the trajectory solves the task. Inferring the latent variable corresponds to learning a dynamics model. Latent variable models are typically learned via an evidence lower bound, and we show how a similar evidence lower bound provides yields a new objective for model-based RL. In the same way that the evidence lower bound is a joint optimization problem over two variables, our objective will *jointly* optimize the model and the policy using the same objective: to produce realistic and high-return trajectories. Our objective differs from standard model-based RL objectives, where it is more common to pit the model *against* the policy [5, 33, 37]. A consequence of maximizing the lower bound is that the dynamics model does not learn the true dynamics, but rather learns optimistic dynamics that facilitate exploration.

The main contribution of this work is an objective for model-based RL. To the best of our knowledge, this is the first lower bound for model-based RL that holds globally (unlike Luo et al. [30]) and can be efficiently estimated in continuous settings (unlike Kearns and Singh [26]). It is the first lower bound that jointly optimizes the model and policy using the same objective. We also present a more complex version of this bound that becomes tight under some assumptions. Through numerical simulations in simple tasks, we demonstrate that optimizing the bound yields reward-maximizing policies and yields an optimistic dynamics model that can aid exploration. We also demonstrate that our bound gracefully accounts for function approximation error in the model – the policy is penalized for taking transitions that the model cannot represent. Finally, we show that we can use parts of our theoretically-motivated objective to design a practically-applicable deep RL method that, despite deviating from the theory, can match the performance of prior model-based methods.

## 2 Related Work

Most model-based RL methods use maximum likelihood to fit the dynamics model and then use RL to maximize the expected return under samples from that model [10, 11, 23, 24, 48]. As noted in prior work, this maximum likelihood objective is not aligned with the RL objective: models that achieve higher likelihood do not necessarily produce better policies [17, 29, 30, 45, 52]. This issue is referred to as the *objective mismatch* problem: the model and policy (or planner) are optimized using different objectives. This problem arises in almost all model-based RL approaches, including those that train the model to predict the value function [34, 41] or that perform planning [10, 41].

Some prior work addresses this problem by decreasing the rewards at states where the model is inaccurate [27, 30, 42, 49, 51], a strategy that our method will also employ through a discriminator. While some of these methods also use discriminators in this manner, ours is the first to optimize a lower bound on returns. Other work modifies the model objective to include multi-step rollouts [3, 4, 17, 25, 45, 47]. Our method will modify the model objective in a different way, so that the model objective is exactly the same as the policy objective. Some prior work directly optimizes the model to produce good policies [2, 13, 32, 35, 43], as theoretically analyzed in Grimm et al. [20]. While our aim is the same as these prior methods, our approach will not require differentiating through unrolled model updates or optimization procedures.

Our work builds on prior lower bounds for model-based RL. Kearns and Singh [26] provide a lower bound that holds globally, but that is only computable in tabular settings. Luo et al. [30] provide a lower bound that can be efficiently estimated, but which only holds for nearby policies and models.

Table 1: **Lower bounds for model-based RL.**

| | Luo et al. [30] | Kearns and Singh [26] | MnM (Eq. 2) | MnM (Eq. 10) |
|---|---|---|---|---|
| holds globally | ✗ | ✓ | ✓ | ✓ |
| efficient to compute | ✓ | ✗ | ✓ | ✓ |
| unified objective | ✗ | ✗ | ✓ | ✓ |
| tight at optimality | ✗[†] | ✓ | ✗ | ✓ |

[†] Discrepancy measure is non-zero in stochastic environments.

As shown in Table 1, our bound combines the strengths of these prior works, providing a lower bound that holds globally and can be efficiently estimated in MDPs with continuous states and actions. Unlike prior work, our bound also mends the objective mismatch problem.

Our theoretical derivation builds on prior work that casts model-based RL as a two-player game between a model-player and a policy-player [5, 33, 36, 38]. Whereas prior work pits model and policy against one another, our formulation will result in a cooperative game: the model and policy cooperate to optimize the *same* objective (a lower bound on the expected return). Our approach, though structurally resembling a GAN, is different from prior work that replaces a maximum likelihood model with a GAN model [6, 8, 28].

The most similar prior work is VMBPO [9], which also jointly optimizes the model and the policy using the same objective. However, while our objective is a lower bound on expected return, VMBPO maximizes a different, risk-seeking objective, which is an *upper* bound on expected return (see Appendix B.1). This different objective can be expressed as the expected return plus the variance of the return, so VMBPO has the undesirable property of preferring policies that receive slight lower return if the variance of the return is much larger (see Appendix B.1). Indeed, while most of the components of our method (e.g., classifiers, GAN-like models) have been used in prior work, our paper is the first to provide a precise recipe for combining these components into an objective that is a provable lower bound.

## 3  A Unified Objective for Model-Based RL

**Notation.**  We focus on the Markov decision process with states $s_t$, actions $a_t$, initial state distribution $p_0(s_0)$, positive reward function $r(s_t, a_t) > 0$ , and dynamics $p(s_{t+1} \mid s_t, a_t)$. Our aim is to learn a control policy $\pi_\theta(a_t \mid s_t)$ with parameters $\theta$ that maximizes the expected discounted return:

$$\max_\theta \mathbb{E}_{\pi_\theta}\left[ \sum_{t=0}^\infty \gamma^t r(s_t, a_t) \right]. \tag{1}$$

We use transitions $(s_t, a_t, r_t, s_{t+1})$ collected from the (real) environment to train the dynamics model $q_\theta(s_{t+1} \mid s_t, a_t)$, and use transitions sampled from this learned model to train the policy. To simplify notation, we will define a trajectory $\tau \triangleq (s_0, a_0, s_1, a_1, \cdots)$ as a sequence of states and actions visited in an episode. We define $R(\tau) \triangleq \sum_{t=0}^\infty \gamma^t r(s_t, a_t)$ as the discounted return of a trajectory. We define two distributions over trajectories: $p^\pi(\tau)$ and $q^\pi(\tau)$ for when policy $\pi_\theta$ interacts with dynamics $p(s_{t+1} \mid s_t, a_t)$ and $q_\theta(s_{t+1} \mid s_t, a_t)$, respectively:

$$p^\pi(\tau) = p_0(s_0) \prod_{t=0}^\infty p(s_{t+1} \mid s_t, a_t)\pi_\theta(a_t \mid s_t), \quad q^\pi(\tau) = p_0(s_0) \prod_{t=0}^\infty q_\theta(s_{t+1} \mid s_t, a_t)\pi_\theta(a_t \mid s_t).$$

**Desiderata.**  Our aim is to design an objective $\mathcal{L}(\theta)$ with two properties. *First*, this objective should be a lower bound on the expected return in the true environment: if the policy does well in the learned model, we are guaranteed that the policy will also do well in the true environment. The expected return under the learned model, which most prior model-based RL methods use to train the policy, is not a lower bound on the expected return [26, 30]. While prior work has made strides in developing lower bounds for model-based RL, even the best lower bounds do not hold for all models [30] or are limited to tabular settings [26].

*Second*, this objective should be the same for the policy and the dynamics model, such that updates to the model would improve the policy, and vice versa. This desiderata is important because prior work has found that training the model to be more accurate (increase likelihood) can decrease the policy's expected return under that model [17, 29, 30, 45, 52]. One prior method (VMBPO [9]) does train the model and policy with the same objective, but this objective corresponds to a risk-seeking, upper-bound on the expected returns.

**An objective for model-based RL.**  We now introduce an objective that achieves these aims. The key idea to deriving this result is to take a probabilistic perspective on decision making: we view the trajectory as an unobserved random variable, and the reward function as the probability that a trajectory solves a task. Then, the problem of inferring this random variable is equivalent to learning the dynamics model. Using an evidence lower bound, we obtain an objective for jointly training the model and policy. We provide the full derivation in Appendix B.3.

Our resulting objective is the policy's reward when interacting with the learned model, but using a new reward function. The new reward function combines the task reward with a term that measures

the difference between the learned model and the real environment. We define our objective

$$\mathcal{L}(\theta) \triangleq \mathbb{E}_{q^{\pi_\theta}(\tau)} \left[ \sum_{t=0}^{\infty} \gamma^t \tilde{r}(s_t, a_t, s_{t+1}) \right],$$  (2)

where the modified reward function is defined as

$$\tilde{r}(s_t, a_t, s_{t+1}) \triangleq (1 - \gamma) \log r(s_t, a_t) + \log \left( \frac{p(s_{t+1} \mid s_t, a_t)}{q(s_{t+1} \mid s_t, a_t)} \right) - (1 - \gamma) \log(1 - \gamma).$$  (3)

Intuitively, the new reward function $\tilde{r}$ penalizes the policy for taking transitions that are unlikely under the true dynamics model, similar to prior work [15, 46, 49]. Later, we will show that we can estimate this augmented reward *without knowing the true environment dynamics* by using a classifier.

We will optimize this lower bound with respect to both the policy $\pi_\theta(a_t \mid s_t)$ and the dynamics model $q_\theta(s_{t+1} \mid s_t, a_t)$. For the policy, we maximize the modified reward using samples from the learned model; the only difference from prior work is the modification to the reward function. Training the dynamics model using this objective is very different from standard maximum likelihood training. The model is optimized to sample trajectories that are similar to real dynamics (like a GAN) and that have high reward (unlike a GAN). This objective differs from VMBPO [9] by taking the $\log(\cdot)$ of the original reward functions; our experiments demonstrate that excluding this component invalidates our lower bound and results in learning suboptimal policies (Fig. 3a).

Our objective achieves the two desiderata. *First*, our objective is a lower bound on the expected return. To state this result formally, we will take the logarithm of the expected return. Of course, maximizing the $\log(\cdot)$ of the expected return is equivalent to maximizing the expected return.

**Theorem 3.1.** *The following bound holds for **any** dynamics $q(s_{t+1} \mid s_t, a_t)$ and policy $\pi(a_t \mid s_t)$:*

$$\log \mathbb{E}_\pi \left[ \sum_{t=0}^{\infty} \gamma^t r(s_t, a_t) \right] \geq \mathcal{L}(\theta).$$

The proof is presented in Appendix B.3. To the best of our knowledge, this is the first global (unlike Luo et al. [30]) and efficiently-computable (unlike Kearns and Singh [26]) lower bound for model-based RL. The model and the policy are trained using the same objective: updating the model not only increases the objective for the model, but also increases the objective for the policy.

Sec. 4 will introduce an algorithm to maximize this lower bound. While this lower bound may not be tight, experiments in Sec. 5 demonstrate that optimizing this lower bound yields policies that achieve high reward across a wide range of tasks. In Appendix A, we propose a variant of this bound that does become tight at convergence. Because this tight bound is more complex, we focus on the simple bound in this paper.

**The optimal dynamics are optimistic.** We now return to analyzing the simpler lower bound ($\mathcal{L}(\theta)$ in Eq. 2). In stochastic environments, the dynamics model that optimizes this lower bound is not equal to the true environment dynamics. Rather, it is biased towards sampling trajectories with high return. Ignoring parametrization constraints, the dynamics model that optimizes our lower bound is $q^*(\tau) = \frac{p(\tau)R(\tau)}{\int p(\tau')R(\tau')d\tau'}$ (proof in Appendix B.4.). While it may seem surprising that the objective-optimizing dynamics would differ from the true dynamics, this result is analogous to a VAE, where the ELBO-optimizing encoder differs from the prior. The optimism in the dynamics model may accelerate policy optimization, a hypothesis we will test in Sec. 5.1.

Would the optimistic dynamics overestimate the policy's return, violating Theorem 3.1? While using the optimistic dynamics with the *original* reward function will overestimate the true return, using the optimistic dynamics with the *augmented* reward function yields a valid lower bound. We demonstrate this effect in Fig. 3b.

## 4 Practical Optimization of the Lower Bound

The previous section presented a single (global) lower bound ($\mathcal{L}$ from Eq. 2) for jointly optimizing the policy and the dynamics model. In this section, we develop a practical algorithm for optimizing

this lower bound. The main challenge in optimizing this bound is that the augmented reward function depends on the transition probabilities of the real environment, $p(s_{t+1} \mid s_t, a_t)$, which are unknown. We address this challenge by learning a classifier (Sec. 4.1). Of course, Theorem 3.1 only holds if the classifier is Bayes-optimal. We then describe the precise update rules for the policy, dynamics model, and classifier (Sec. 4.2).

## 4.1 Estimating the Augmented Reward Function

To estimate the augmented reward function, which depends on the transition probabilities of the real environment, we learn a classifier that distinguishes real transitions from fake transitions. This approach is similar to GANs [19] and similar to prior work in RL [15, 49]. We use $C_\phi(s_t, a_t, s_{t+1}) \in [0, 1]$ to denote the classifier, which we train to distinguish real versus model transitions using the standard cross entropy loss:

$$\max_\phi \mathcal{L}_C(s_t^{\text{real}}, a_t^{\text{real}}, s_{t+1}^{\text{real}}, s_{t+1}^{\text{model}}; \phi) \triangleq \log C_\phi(s_t^{\text{real}}, a_t^{\text{real}}, s_{t+1}^{\text{real}}) + \log\left(1 - C_\phi(s_t^{\text{real}}, a_t^{\text{real}}, s_{t+1}^{\text{model}})\right).$$
(4)

Note that the real transition $(s_t^{\text{real}}, a_t^{\text{real}}, s_{t+1}^{\text{real}})$ and model transition $(s_t^{\text{real}}, a_t^{\text{real}}, s_{t+1}^{\text{model}})$ have the same initial state and initial action. Once trained, we can use the classifier's predictions to estimate the augmented reward function:

$$\tilde{r}(s_t, a_t, s_{t+1}) \approx \log r(s_t, a_t) + \log\left(\frac{C_\phi(s_t, a_t, s_{t+1})}{1 - C_\phi(s_t, a_t, s_{t+1})}\right).$$
(5)

The approximation above reflects function approximation error in learning the classifier. Now that we can estimate the augmented reward function, we can apply *any* RL method to maximize the augmented reward under transitions sampled from the dynamics model. The following section describes a particular instantiation using an off-policy RL algorithm.

## 4.2 Updating the Model and Policy

We now present our complete method, which trains three components: a classifier, a policy, and a dynamics model. Our method alternates between *(1)* updating the policy (by performing RL using model experience with augmented rewards) and *(2)* updating the dynamics model and classifier (using a GAN-like objective). In describing the loss functions below, we use the superscripts $(\cdot)^{\text{real}}$ and $(\cdot)^{\text{model}}$ to denote transitions that have been sampled from the true environment dynamics or the learned dynamics function. To reduce clutter, we omit the superscripts when unambiguous.

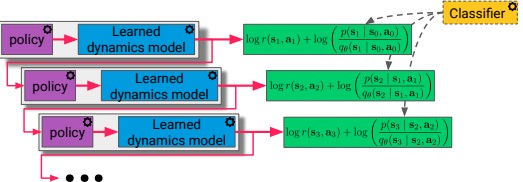

Figure 1: **Mismatched No More** is a model-based RL algorithm that learns a policy, dynamics model, and classifier. The classifier distinguishes real transitions from model transitions. The policy and dynamics are jointly optimized to sample transitions that yield high returns and look realistic, as estimated by the classifier.

**Updating the policy.** The policy is optimized to maximize the augmented reward on transitions sampled from the learned dynamics model. While this optimization can be done using any RL algorithm, including on-policy methods, we will focus on an off-policy actor-critic method.

We define the Q function as sum of *augmented* rewards under the learned dynamics model:

$$Q(s_t, a_t) \triangleq \mathbb{E}_{\substack{\pi(a_t \mid s_t), \\ q_\theta(s_{t+1} \mid s_t, a_t)}} \left[ \sum_{t'=t}^{\infty} \gamma^{t'-t} \tilde{r}(s_{t'}, a_{t'}) \mid {\substack{s_t = s_t, \\ a_t = a_t}} \right].$$
(6)

We approximate the Q function using a neural network $Q_\psi(s_t, a_t)$ with parameters $\phi$. We train the Q function using the TD loss on transitions sampled from the *learned* dynamics model:

$$\mathcal{L}_Q(s_t, a_t, r_t, s_{t+1}^{\text{model}}; \psi) = \left(Q_\psi(s_t, a_t) - \lfloor y_t \rfloor_{\text{sg}}\right)^2,$$
(7)

where $\lfloor \cdot \rfloor_{\text{sg}}$ is the stop-gradient operator and $y_t = \tilde{r}(s_t, a_t, s_{t+1}^{\text{model}}) + \gamma \mathbb{E}_{\pi(a_{t+1} \mid s_{t+1}^{\text{model}})} \left[ Q_\psi(s_{t+1}^{\text{model}}, a_{t+1}) \right]$ is the TD target. The augmented reward $\tilde{r}$ is estimated using the learned classifier (Eq. 5). To estimate

**Algorithm 1 Mismatched no More (MnM)** is an algorithm for model-based RL. The method alternates between training the policy on experience from the learned dynamics model with augmented rewards and updating the model+classifier using a GAN-like loss. While we use an off-policy RL algorithm on L4, any other RL algorithm can be substituted.

---

1: **while** not converged **do**
2:     Sample experience from the learned model.
3:     Modify rewards using the classifier (Eq. 5).
4:     Update the policy and Q function using the model experience and modified rewards (Eq.s 8 and 7).
5:     Update model and classifier using GAN-like losses (Eq.s 4 and 9).
6:     (Infrequently) Sample experience from the real model.
7: **return** policy $\pi_\theta(a_t \mid s_t)$.

---

the corresponding value function, we use a 1-sample approximation: $V_\psi(s_t) = Q_\psi(s_t, a_t)$ where $a_t \sim \pi_\theta(a_t \mid s_t)$). The policy is trained to maximize the Q function:

$$\max_\theta \mathcal{L}_\pi(s_t; \theta) \triangleq \mathbb{E}_{\pi_\theta(a_t|s_t)} \left[ Q_\psi(s_t, a_t) \right]. \tag{8}$$

In our implementation, we regularize the policy by adding an additional entropy regularizer. Following prior work [18], we maintain two Q functions and two target Q functions, using the minimum of the two to compute the TD target. See Appendix D for details.

**Updating the dynamics model.** To optimize the dynamics model, we rewrite the lower bound in terms of a single transition (derivation in Appendix B.6):

$$\mathcal{L}_q(s_t^{\text{real}}, a_t^{\text{real}}; \theta) = \mathbb{E}_{s_{t+1}^{\text{model}} \sim q_\theta(s_{t+1}|s_t^{\text{real}}, a_t^{\text{real}})} \left[ V_\psi(s_{t+1}^{\text{model}}) + \log \left( \frac{C_\phi(s_t^{\text{real}}, a_t^{\text{real}}, s_{t+1}^{\text{model}})}{1 - C_\phi(s_t^{\text{real}}, a_t^{\text{real}}, s_{t+1}^{\text{model}})} \right) \right]. \tag{9}$$

This loss is an approximation of our original lower bound (Eq. 2) because we estimate the difference in dynamics using a classifier. This approximation is standard in prior work on GANs [19] and adversarial inference [12, 14].

Intuitively, the procedure for optimizing the dynamics model and the classifier resembles a GAN [19]: the classifier is optimized to distinguish real transitions from model transitions, and the model is updated to fool the classifier (and increase rewards). However, *our method is not equivalent to simply replacing a maximum likelihood model with a GAN model*. Indeed, such an approach would not optimize a lower bound on expected return. Rather, our model objective includes an additional value term and our policy objective includes an additional classifier term. These changes enable the model and policy to optimize the same objective, which is a lower bound on expected return.

**Algorithm summary.** We summarize the method in Alg. 1 and provide an illustration in Fig. 1. We call the method MISMATCHED NO MORE (MnM) because the policy and model optimize the same objective, thereby resolving the objective mismatch problem noted in prior work. While the model and policy are optimized using the same objective, that objective can stop being a lower bound if the learned classifier is not Bayes-optimal.

Implementing MnM on top of a standard model-based RL algorithm is straightforward. First, create an additional classifier network. Second, instead of using the maximum likelihood objective to train the model, use the GAN-like objective in Eq. 9 to update both the model and the classifier. Third, add the classifier's logits to the predicted rewards (Eq. 5). Following prior work [24], we learn a neural network to predict the true environment rewards.

### 4.3 A Note about Exploration

The classifier term in the augmented reward (Eq. 3) is a double edged sword. Our theoretical derivation suggests that this term should appear, and our didactic experiments demonstrate that removing this term results in suboptimal behavior. Experiments also show that this term can effectively combat errors in the learned model. Prior work in offline RL has found similar model-error reward terms critical for achieving good performance in the offline setting [27, 42, 49, 51].

However, when scaling MnM to continuous control tasks in the online setting, we found that including this term hurts performance (Fig. 10). This makes sense: this classifier term is exactly the opposite

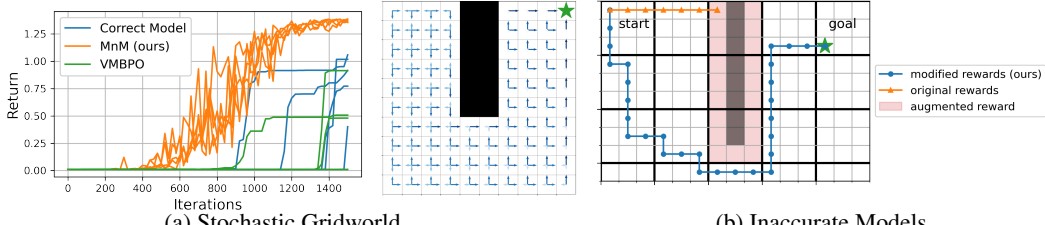

(a) Stochastic Gridworld          (b) Inaccurate Models

Figure 2: **Gridworld experiments.** *(Left)* We apply MnM to a navigation task with transition noise that moves the agent to neighboring states with equal probability. MnM solves this task more quickly than Q-learning and VMBPO. The dynamics learned by MnM are different from the real dynamics, changing the transition noise (blue arrows) to point towards the goal. *(Right)* We simulate function approximation by a learning model that makes the same predictions for groups of $3 \times 3$ states, resulting in a model that is inaccurate around obstacles. The modified reward compensates for these errors by penalizing the policy for navigating near obstacles.

of exploration objectives based on model error (e.g., [44]), and penalizes the policy for performing exploration. Our continuous control experiments in Sec. 5.2 will deviate from our theoretical derivation: they will use the model objective suggested by our theory, but the same policy objective as prior. That is, the policy maximizes $r(s_t, a_t)$ instead of $\tilde{r}(s_t, a_t)$ (Eq. 3). We call this method "MnM-approx." We note that many theoretical model-based RL papers also find it necessary to implement practical algorithms that differ from the theory [30, 46, 49].

## 5 Numerical Simulations

The primary aim of our experiments is to verify that our objective is a valid lower bound, and that maximizing the bound yields reward-maximizing policies. We study these questions in Sec. 5.1, where we will use tabular problems so that we can study the objective in the absence of function approximation error. Then, in Sec. 5.2, we show adapting part of our objective into a scalable model-based RL algorithm yields a method that, while deviating from the theory, can achieve competitive results on benchmark tasks.

### 5.1 Understanding the Lower Bound and the Learned Dynamics

Our didactic experiments test whether optimizing the lower bound produces optimal policies and study how the components of the lower bound. We use tabular domains in this section, as they allow us to analytically compute the optimal policy for comparison, and allow us to evaluate the lower bound in the absence of function approximation error.

Our first experiment compares the lower bound to an oracle method that applies Q-learning to a perfect dynamics model. In comparison to this baseline, MnM learns a dynamics model using the GAN-like objective (Eq. 9) and maximizes a modified reward function (Eq. 5). We also compare to VMBPO, which learns a dynamics model similar to MnM but omits the log-transformation of the reward function; this transformation ordinarily encourages pessimistic behavior. For this experiment, we use a $10 \times 10$ gridworld with stochastic dynamics and sparse reward, shown in Fig. 2a *(Left)*. The results, shown in Fig. 2a, show that MnM outperforms both Q-learning with the correct model and VMBPO on this task. We hypothesize that VMBPO performs poorly on this task because it maximizes an upper bound on performance; and confirm this in the following experiments.

We hypothesize that MnM outperforms Q-learning with the correct model because it learns "optimistic" dynamics. We test this hypothesis by visualizing the dynamics model learned by MnM (Fig. 2a *(Left)*). While the true environment dynamics have stochasticity that moves the agent in a random direction with *equal* probability, the MnM dynamics model biases this stochasticity to lead the agent towards the goal (blue arrows point towards the goal). Of course, we use the true environment dynamics, not the optimistic dynamics model, for evaluating the policies.

Our augmented reward function contains two crucial components, *(1)* the classifier term and *(2)* the logarithmic transformation of the reward function. We test the importance of the classifier term in correcting for inaccurate models. To do this, we limit the capacity of the MnM dynamics model so that it makes "low-resolution" predictions, forcing all states in $3 \times 3$ blocks to have the same dynamics. We will use the gridworld shown in Fig. 2b *(Left)*, which contains obstacles that occur

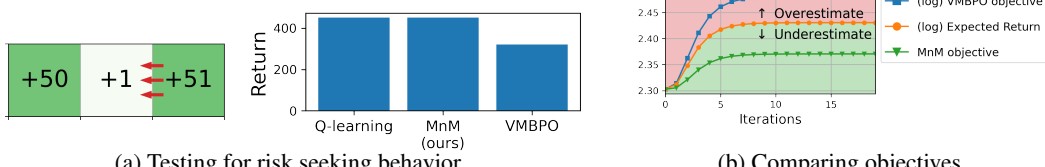

|                          |                        |
|--------------------------|------------------------|
| (a) Testing for risk seeking behavior | (b) Comparing objectives. |

Figure 3: **Analyzing the lower bound.** *(Left)* On a 3-state MDP with stochastic transitions in one state (red arrows), MnM learns the reward-maximizing policy while VMBPO learns a strategy with lower rewards and higher variance (as predicted by theory). *(Right)* We apply value iteration to the gridworld from Fig. 2a to analytically compute various objectives. As predicted by our theory, the MnM objective is a lower bound on the expected return, whereas the VMBPO objective overestimates the expected return.

at a finer resolution than the model can detect. When the "low resolution" dynamics model makes predictions for states near the obstacle, it will average together some states with obstacles and some states without obstacles. Thus, the model will (incorrectly) predict that the agent always has some probability of moving in each direction, even if that direction is actually blocked by an obstacle. However, the classifier (whose capacity we have also limited) detects that the dynamics model is inaccurate in these states, so the augmented reward is much lower at these states. Thus, MnM is able to solve this task despite the inaccurate model; an ablation of MnM that removes the classifier term attempts to navigate through the wall and fails to reach the goal.

Like MnM, VMBPO includes a classifier term in the reward function but omits the logarithmic transformation, a difference we expect to cause VMBPO to prefer suboptimal, risk-seeking policies. To test this hypothesis, we use the 3-state MDP in Fig. 3a *(Right)*, where numbers indicate the reward at each state. While moving to the right state yields slightly higher rewards, "wind" knocks the agent out of this state with probability 50% so the reward-maximizing strategy is to move to the left state. While MnM learns the reward-maximizing strategy, VMBPO learns a policy that goes to the right state and receives lower returns.

Finally, we verify Theorem 3.1 by comparing the MnM objective to the true expected return. We also compare to the objective from VMBPO, which looks similar to the MnM objective but omits the logarithmic transformation; our theory predicts that the VMBPO objective will therefore be an *upper* bound on the expected return (see Appendix B.1). We use the gridworld from Fig. 2a and use a version of MnM based on value iteration to avoid approximation error. Plotting the MnM objective in Fig. 3b *(Right)*, we observe that it is always a lower bound on the (log) expected return, as predicted by our theory. Also as predicted by the theory, the VMBPO objective overestimates the expected return, illustrating the importance of the logarithmic transformation.

Of the oft-cited benefits of model-based RL is that the learned dynamics model can be re-used to solve new tasks. MnM presents a twist on that story, because MnM does not learn the true environment dynamics but rather learns optimistic dynamics (see Sec. 3). In Appendix C (Fig. 13), we examine MnM's effectiveness at transferring dynamics to different tasks in the stochastic gridworld from Fig. 2a. We find that the (optimistic) dynamics learned by MnM do not slow learning on dissimilar tasks, but can accelerate learning of challenging, similar tasks.

## 5.2 Comparisons On Higher-Dimensional Tasks

Our next experiments use continuous-control robotic tasks to study whether the new model-learning objective suggested by our lower bound can be stably applied to higher-dimensional control tasks. While the MnM-approx method tested in this section deviates from our theoretical derivation (see Sec. 4.3), these experiments are nonetheless useful for providing preliminary evidence about whether the proposed model objective can be efficiently estimated and stably optimized.

We use MBPO [24] as a baseline for model-based RL because it achieves state-of-the-art results and is a prototypical example of model-based RL algorithms that use maximum likelihood models. Because our method differs from the baseline (MBPO) along only one dimension now (new model update, same policy update), these experiments will directly test the utility of our proposed model update. Experimental details are in Appendix D.

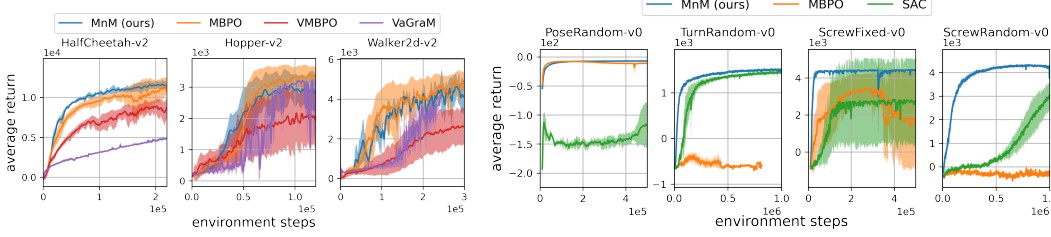

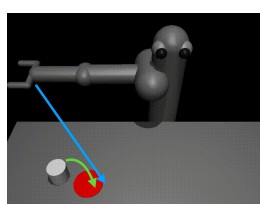
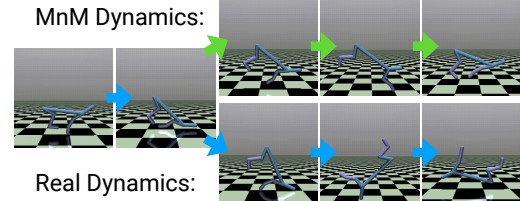

(a) OpenAI Gym benchmark           (b) ROBEL manipulation benchmark

Figure 4: **Comparison on two benchmarks**. *(Left)* On the OpenAI gym benchmark [7], MnM-approx performs on par with a prior state-of-the-art method (MBPO [24]), while consistently outperforming a recent method that addresses objective mismatch (VMBPO [9]). *(Right)* The ROBEL manipulation benchmark [1] contains complex contact dynamics that are challenging to model. MBPO performs poorly on these tasks, often worse than model-free SAC [22].

Figure 5: **Optimistic Dynamics**: *(Left)* On the `Pusher-v2` task, the MnM dynamics model makes the puck move towards the puck move towards the gripper before being grasped. *(Right)* On the `HalfCheetah-v2` task, the MnM dynamics model helps the agent stay upright after tripping.

We first use three locomotion tasks from the OpenAI Gym benchmark [7] to compare MnM-approx to MBPO and VMBPO. The VMBPO curves are taken directly from that paper. As shown in Fig. 4a, MnM-approx performs roughly on par with MBPO and outperforms [9], a more recent model-based method also addresses the objective mismatch problem by maximizing an *upper* bound on expected returns (1-sided p-values: $p = 0.04, 0.00, 0.03$).

We next use the ROBEL manipulation benchmark [1] to compare how MnM-approx and MBPO handle tasks with more complicated dynamics. As shown in Fig. 4b, MBPO struggles to learn three of the four tasks, likely because the dynamics are hard. In contrast, MnM-approx solves these tasks, likely because the GAN-like model is more accurate. MnM-approx outperforms MBPO on all tasks ($p \leq 0.03$) and outperforms the model-free baseline on 2/4 tasks ($p \leq 0.02$)

We next compare to prior methods for addressing the objective mismatch problem using the `DClawScrewFixed-v0` task. We compare to VAML [17] and the value-weighted maximum likelihood approach proposed in Lambert et al. [29]. As shown in Fig. 6, these alternative model learning objectives perform poorly on this task.

To better understand why MnM-approx sometimes outperforms the maximum likelihood baseline (MBPO), we visualized the Q-values throughout training. We used `metaworld-drawer-open-v2`, a task where we found a noticeable difference in the performance between MBPO and MnM-approx. Fig. 7a shows that MnM-approx yields Q values that are more accurate and more stable than MBPO, perhaps because MBPO learns a policy that exploits inaccuracies in the learned model.

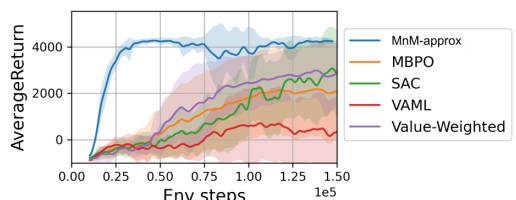

Figure 6: **Alternative model learning objectives**: Using the `DClawScrewFixed-v0` task, we compare MnM-approx and MBPO [24] to two additional model learning objectives suggested in the literature, VAML [17] and value-weighted maximum likelihood [29]. MnM-approx outperforms these alternative approaches.

To study the stability of MnM-approx, we plot the validation MSE of the MnM-approx model when training on the `DClawScrewRandom-v0` task. As shown in Fig. 7b, the MSE decreases stably, indicating that the adversarial nature of the MnM-approx model training does not create instabilities. In Appendix C (Fig. 8, 9) we visualize Q

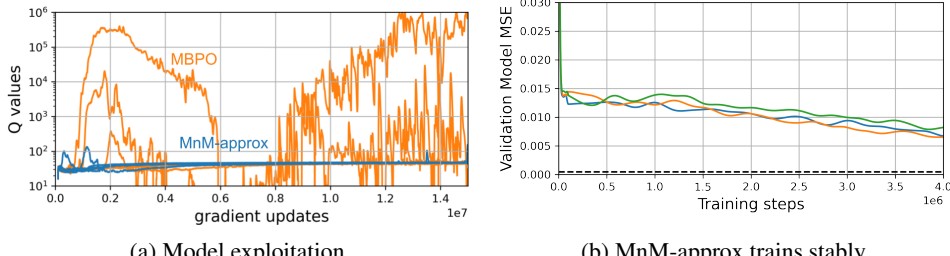

| (a) Model exploitation | (b) MnM-approx trains stably. |

**Figure 7: Analyzing MnM-approx.** *(Left)* The very large Q values of MBPO suggest model exploitation, which our method appears to avoid. *(Right)* Despite resembling a GAN, the MnM-approx dynamics model trains stably. Different colors correspond to different random seeds, and the dashed line corresponds to the minimum validation MSE of an MLE dynamics model.

value stability and MSE model error on additional environments. In Appendix C (Fig. 12), we study the effect of model horizon on MnM in a continuous control task, and find that performance degrades linearly as we increase this value.

We visualize the dynamics learned by MnM on two robotic control tasks in Fig 5. These tasks have deterministic dynamics, so our theory would predict that an idealized version of MnM would learn a dynamics model exactly equal to the deterministic dynamics. However, our implementation relies on function approximation (neural networks) to learn the dynamics, and the limited capacity of function approximators makes otherwise-deterministic dynamics appear stochastic. On the `Pusher-v2` task, the MnM dynamics cause the puck to move towards the robot arm even before the arm has come in contact with the puck. While this movement is not physically realistic, it may make the exploration problem easier. On the `HalfCheetah-v2` task, the MnM dynamics increase the probability that the agent remains upright after tripping, likely making it easier for the agent to learn how to run. We expect that the implicit stochasticity caused by function approximation to be especially important for real-world tasks, where the complexity of the real dynamics often dwarfs the capacity of the learned dynamics model.

## 6 Conclusion

The main contribution of this paper is an objective for model-based RL that is both efficient to compute and globally valid. Moreover, the model and policy are jointly optimized using this same objective, thereby addressing the objective mismatch problem. This *joint optimization* will ease and accelerate the design of future model-based RL algorithms.

The main limitation of this paper is the classifier term. Because this term is estimated, the objective used in practice can fail to be a lower bound when the classifier is not Bayes-optimal. Additionally, this classifier term can hinder exploration and degrade performance on continuous control tasks in the online setting. We encourage future investigations into *joint optimization* based approaches that do explicitly address the role of exploration, perhaps including the dataset as an additional optimization variable.

**Acknowledgements.** We thank Laura Smith, Marvin Zhang, and Dibya Ghosh for helpful discussions, and thank Anusha Nagabandi and Michael Janner for help with experiments. We thank Yinlam Chow for help sharing the VMBPO baselines and thank Danijar Hafner for reviewing a draft of the paper. This material is supported by the Fannie and John Hertz Foundation and the NSF GRFP (DGE1745016).

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
