# A    Tightening the lower bound

We now introduce a modification to our lower bound that does make the bound tight. This new lower bound will be more complex than the one introduced above and we have not yet successfully designed an algorithm for maximizing it. Nonetheless, we believe that presenting the bound may prove useful for the design of future model-based RL algorithms.

We will use $\mathcal{L}_\gamma(\theta)$ to denote this new lower bound. In addition to the policy and dynamics, this bound will also depend on a time-varying discount, $\gamma_\theta(t)$, in place of the typical $\gamma^t$ term. Similar learned discount factors have been studied in previous work on model-free RL [39]. We define this objective as follows:

$$\mathcal{L}_\gamma(\theta) \triangleq \mathbb{E}_{q^{\pi_\theta}(\tau)}\left[\sum_{t=0}^{\infty} \gamma_\theta(t)\tilde{r}_\gamma(s_t, a_t, s_{t+1})\right], \tag{10}$$

where the augmented reward is now defined as

$$\tilde{r}_\gamma(s_t, a_t, s_{t+1}) \triangleq \log r(s_t, a_t) + \log\left(\frac{\gamma^t}{\gamma_\theta(t)}\right)$$

$$+ \frac{1 - \Gamma_\theta(t-1)}{\gamma_\theta(t)} \log\left(\frac{p(s_{t+1} \mid s_t, a_t, s_{t-1}, a_{t-1}, \cdots)}{q_\theta(s_{t+1} \mid s_t, a_t, s_{t-1}, a_{t-1}, \cdots)}\right),$$

and $\Gamma_\theta(t) = \sum_{t'=0}^{t} \gamma_\theta(t')$ is the CDF of the learned discount function (i.e., $\gamma_\theta(t)$ is a probability distribution over $t$.). This new lower bound, which differs from our main lower bound by the learnable discount factor, does provide a tight bound on the expected return objective:

**Lemma A.1.** *Let an arbitrary policy $\pi(a_t \mid s_t)$ be given. The objective $\mathcal{L}_\gamma(\theta)$ is also a lower bound on the expected return objective, $\log \mathbb{E}_\pi \left[\sum_{t=0}^{\infty} \gamma^t r(s_t, a_t)\right] \geq \mathcal{L}_\gamma(\theta)$, and this bound becomes tight at optimality:*

$$\log \mathbb{E}_\pi \left[\sum_{t=0}^{\infty} \gamma^t r(s_t, a_t)\right] = \max_{q^\pi(\tau), \gamma_\theta(t)} \mathcal{L}_\gamma(\theta).$$

The proof is presented in Appendix B.4. One important limitation of this result is that the learned dynamics that maximize this lower bound to make the bound tight may be non-Markovian. Intriguingly, this analysis suggests that using non-Markovian models, such as RNNs and transformers, may accelerate learning on Markovian tasks. This paper does not propose an algorithm for optimizing this more complex lower bound.

# B    Proofs and Additional Analysis

## B.1    VMBPO Maximizes an Upper Bound on Return

While MnM aims to maximize the (log) of the expected return, VMBPO aims to maximize the expected *exponentiated* return:

$$\text{MnM:} \quad \log \mathbb{E}_\pi\left[\sum_{t=0}^{\infty} \gamma^t r(s_t, a_t)\right], \qquad \text{VMBPO:} \quad \log \mathbb{E}_\pi\left[e^{\eta \sum_{t=0}^{\infty} \gamma^t r(s_t, a_t)}\right],$$

where $\eta > 0$ is a temperature term used by VMBPO. Note that maximizing the log of the expected return, as done by MnM, is equivalent to maximizing the expected return, as the function $\log(\cdot)$ is monotone increasing. However, maximizing the log of the expected *exponentiated* return, as done by VMBPO, is not equivalent to maximizing the expected return. Rather, it corresponds to maximizing a sum of the expected return and the *variance* of the return [31, Page 272]:

$$\frac{1}{\eta}\log \mathbb{E}_\pi\left[e^{\eta \sum_{t=0}^{\infty} \gamma^t r(s_t, a_t)}\right] = \mathbb{E}_\pi\left[\sum_{t=0}^{\infty} \gamma^t r(s_t, a_t)\right] + \frac{\eta}{2}\text{Var}_\pi\left[\sum_{t=0}^{\infty} \gamma^t r(s_t, a_t)\right] + \mathcal{O}(\eta^2).$$

Thus, in environments with stochastic dynamics or rewards (e.g., the didactic example in Fig. 3a), VMBPO will prefer to receive lower returns if the variance of the returns is much higher. We note that the expected *exponentiated* return is an *upper* bound on the expected return:

$$\log \mathbb{E}_\pi\left[e^{\eta \sum_{t=0}^{\infty} \gamma^t r(s_t, a_t)}\right] \geq \eta \mathbb{E}_\pi\left[\sum_{t=0}^{\infty} \gamma^t r(s_t, a_t)\right].$$

This statement is a direct application of Jensen's inequality. The bound holds with a strict inequality in almost all MDPs. The one exception is trivial MDPs where all trajectories have exactly the same return. Of course, even a random policy is optimal for these trivial MDPs.

## B.2 Helper Lemmas

We start by introducing a simple identity that will help handle discount factors in our analysis.

**Lemma B.1.** *Define $p(H) = \text{GEOM}(1 - \gamma)$ as the geometric distribution. Let discount factor $\gamma \in (0, 1)$ and random variable $x_t$ be given. Then the following identity holds:*

$$\mathbb{E}_{p(H)} \left[ \sum_{t=0}^{H} x_t \right] = \sum_{t=0}^{\infty} \gamma^t x_t.$$

The proof involves substituting the definition of the Geometric distribution and then rearranging terms.

*Proof.*

$$\begin{aligned}
\mathbb{E}_{p(H)} \left[ \sum_{t=0}^{H} x_t \right] &= (1 - \gamma) \sum_{H=0}^{\infty} \gamma^H \sum_{t=0}^{H} x_t \\
&= (1 - \gamma) \left( x_0 + \gamma(x_0 + x_1) + \gamma^2(x_0 + x_1 + x_2) + \cdots \right) \\
&= (1 - \gamma) \left( x_0(1 + \gamma + \gamma^2 + \cdots) + x_1(\gamma + \gamma^2 + \cdots) + \cdots \right) \\
&= (1 - \gamma) \left( x_0 \frac{1}{1 - \gamma} + x_1 \frac{\gamma}{1 - \gamma} + x_2 \frac{\gamma^2}{1 - \gamma} + \cdots \right) \\
&= \sum_{t=0}^{\infty} \gamma^t x_t.
\end{aligned}$$

$\square$

The second helper lemma describes how the discounted expected return objective can be written as the expected *terminal* reward of a mixture of finite-length episodes.

**Lemma B.2.** *Define $p(H) = \text{GEOM}(1 - \gamma)$ as the geometric distribution, and $p(\tau \mid H)$ as a distribution over length-$H$ episodes. We can then write the expected discounted return objective as follows:*

$$\mathbb{E}_{p(\tau|H=\infty)} \left[ \sum_{t=0}^{\infty} \gamma^t r(s_t, a_t) \right] = \frac{1}{1 - \gamma} \mathbb{E}_{p(H)} \left[ \mathbb{E}_{p(\tau|H=H)} \left[ r(s_H, a_H) \right] \right] \tag{11}$$

$$= \frac{1}{1 - \gamma} \iint p(H) p(\tau \mid H = H) r(s_H, a_H) d\tau dH. \tag{12}$$

*Proof.* The first identity follows from the definition of the geometric distribution. The second identity writes the expectations as integrals, which will make future analysis clearer. $\square$

## B.3 Proof of Theorem 3.1

*Proof.*

$$\log \mathbb{E}_\pi \left[ \sum_{t=0}^\infty \gamma^t r(s_t, a_t) \right] \overset{(a)}{=} \log \frac{1}{1-\gamma} \iint p(H) p(\tau \mid H = H) r(s_H, a_H) d\tau dH$$

$$= \log \iint p(H) \frac{p(\tau \mid H = H)}{q_\theta(\tau \mid H = H)} q_\theta(\tau \mid H = H) r(s_H, a_H) d\tau dH - \log(1-\gamma)$$

$$\overset{(b)}{\geq} \int p(H) \left( \log \int \frac{p(\tau \mid H = H)}{q_\theta(\tau \mid H = H)} q_\theta(\tau \mid H = H) r(s_H, a_H) d\tau \right) dH - \log(1-\gamma)$$

$$\overset{(c)}{\geq} \iint p(H) q_\theta(\tau \mid H = H) \left( \log p(\tau \mid H = H) - \log q_\theta(\tau \mid H = H) + \log r(s_H, a_H) \right) d\tau dH - \log(1-\gamma)$$

$$\overset{(d)}{=} \iint p(H) q_\theta(\tau \mid H = H) \left( \left( \sum_{t=0}^H \log p(s_{t+1} \mid s_t, a_t) + \underline{\log \pi_\theta(a_t \mid s_t)} - \log q_\theta(s_{t+1} \mid s_t, a_t) - \underline{\log \pi_\theta(a_t \mid s_t)} \right) + \log r(s_H, a_H) \right) d\tau dH$$
$$- \log(1-\gamma)$$

$$\overset{(e)}{=} \iint p(H) q_\theta(\tau \mid H = \infty) \left( \left( \sum_{t=0}^H \log p(s_{t+1} \mid s_t, a_t) - \log q_\theta(s_{t+1} \mid s_t, a_t) \right) + \log r(s_H, a_H) \right) d\tau dH - \log(1-\gamma)$$

$$\overset{(f)}{=} \int q_\theta(\tau) \int p(H) \left( \left( \sum_{t=0}^H \log p(s_{t+1} \mid s_t, a_t) - \log q_\theta(s_{t+1} \mid s_t, a_t) \right) + \log r(s_H, a_H) \right) dH d\tau - \log(1-\gamma)$$

$$\overset{(g)}{=} \int q_\theta(\tau) \mathbb{E}_{p(H)} \left[ \left( \sum_{t=0}^H \log p(s_{t+1} \mid s_t, a_t) - \log q_\theta(s_{t+1} \mid s_t, a_t) \right) + \log r(s_H, a_H) \right] d\tau - \log(1-\gamma)$$

$$\overset{(h)}{=} \int q_\theta(\tau) \sum_{t=0}^\infty \gamma^t \left( \log p(s_{t+1} \mid s_t, a_t) - \log q_\theta(s_{t+1} \mid s_t, a_t) + (1-\gamma) \log r(s_H, a_H) \right) d\tau - \log(1-\gamma)$$

$$\overset{(i)}{=} \mathbb{E}_{q_\theta(\tau)} \left[ \sum_{t=0}^\infty \gamma^t \left( \log p(s_{t+1} \mid s_t, a_t) - \log q_\theta(s_{t+1} \mid s_t, a_t) + (1-\gamma) \log r(s_H, a_H) - (1-\gamma) \log(1-\gamma) \right) \right].$$

For *(a)*, we applied Lemma B.2. For *(b)*, we applied Jensen's inequality. For *(c)*, we applied Jensen's inequality again. For *(d)*, we substituted the definitions of $p_\theta(\tau \mid H)$ and $q_\theta(\tau \mid H)$. For *(e)*, we noted that the term inside the summation only depends on the first $H$ steps of the trajectory, so collecting longer trajectories will not change the result. This allows us to rewrite the integral as an expectation using a single infinite-length trajectory. For *(f)*, we recalled the definition $q_\theta(\tau) = q_\theta(\tau = H = \infty)$ and swap the order of integration. For *(g)*, we express the inner integral over $p(H)$ as an expectation. For *(h)*, we applied the identity from Lemma B.1. For *(i)*, we moved the constant $\log(1-\gamma)$ back inside the integral and rewrote the integral as an expectation. We have thus obtained the desired result. □

## B.4 Proof of Lemma A.1

Before presenting the proof of Theorem 3.1 itself, we show how we derived the lower bound in this more general case. While this step is not required for the proof, we include it because it sheds light on how similar lower bounds might be derived for other problems. We define $\gamma_\theta(H)$ to be a learned distribution over horizons $H$. We then proceed, following many of the same steps as for the proof of Theorem 3.1.

$$\log \mathbb{E}_\pi \left[ \sum_{t=0}^\infty \gamma^t r(s_t, a_t) \right] \overset{(a)}{=} \log \iint \frac{p(\tau, H)}{q_\theta(\tau, H)} q_\theta(\tau, H) r(s_H, a_H) d\tau dH - \log(1 - \gamma)$$

$$\overset{(b)}{\geq} \iint q_\theta(\tau, H) \left( \log p(\tau, H) - \log q_\theta(\tau, H) + \log r(s_H, a_H) d\tau \right) dH - \log(1 - \gamma) \tag{13}$$

$$\overset{(c)}{=} \int \sum_{H=0}^\infty \gamma_\theta(H) q_\theta(\tau \mid H) \left( \left( \sum_{t=0}^H \log p(s_{t+1} \mid s_t, a_t) - \log q_\theta(s_{t+1} \mid s_t, a_t) \right) + \log p(H) - \log \gamma_\theta(H) + \log r(s_H, a_H) d\tau \right) - \log(1 - \gamma)$$

$$\overset{(d)}{=} \int q_\theta(\tau \mid H = \infty) \sum_{H=0}^\infty \gamma_\theta(H) \left( \left( \sum_{t=0}^H \log p(s_{t+1} \mid s_t, a_t) - \log q_\theta(s_{t+1} \mid s_t, a_t) \right) + \log p(H) - \log \gamma_\theta(H) + \log r(s_H, a_H) d\tau \right) - \log(1 - \gamma)$$

$$\overset{(e)}{=} \int q_\theta(\tau) \sum_{H=0}^\infty \gamma_\theta(H) \left( \left( \sum_{t=0}^H \log p(s_{t+1} \mid s_t, a_t) - \log q_\theta(s_{t+1} \mid s_t, a_t) \right) + \log p(H) - \log \gamma_\theta(H) + \log r(s_H, a_H) d\tau \right) - \log(1 - \gamma)$$

$$\overset{(f)}{=} \mathbb{E}_{q_\theta(\tau)} \left[ \sum_{H=0}^\infty \gamma_\theta(H) \left( \left( \sum_{t=0}^H \log p(s_{t+1} \mid s_t, a_t) - \log q_\theta(s_{t+1} \mid s_t, a_t) \right) + \underline{\log(1-\gamma)} + H \log \gamma - \log \gamma_\theta(H) + \log r(s_H, a_H) \right) \right] - \underline{\log(1-\gamma)}$$

$$\overset{(g)}{=} \mathbb{E}_{q_\theta(\tau)} \left[ \sum_{H=0}^\infty \left( \sum_{t=H}^\infty q(t) \right) (\log p(s_{H+1} \mid s_H, a_H) - \log q_\theta(s_{H+1} \mid s_H, a_H)) + \gamma_\theta(H) (H \log \gamma - \log \gamma_\theta(H) + \log r(s_H, a_H)) \right]$$

$$\overset{(h)}{=} \mathbb{E}_{q_\theta(\tau)} \left[ \sum_{H=0}^\infty \left( 1 - \sum_{t=0}^{H-1} q(t) \right) (\log p(s_{H+1} \mid s_H, a_H) - \log q_\theta(s_{H+1} \mid s_H, a_H)) + \gamma_\theta(H) (H \log \gamma - \log \gamma_\theta(H) + \log r(s_H, a_H)) \right]$$

$$\overset{(i)}{=} \mathbb{E}_{q_\theta(\tau)} \left[ \sum_{H=0}^\infty (1 - \Gamma_\theta(H-1)) (\log p(s_{H+1} \mid s_H, a_H) - \log q_\theta(s_{H+1} \mid s_H, a_H)) + \gamma_\theta(H) (H \log \gamma - \log \gamma_\theta(H) + \log r(s_H, a_H)) \right]$$

$$\overset{(j)}{=} \mathbb{E}_{q_\theta(\tau)} \left[ \sum_{H=0}^\infty \gamma_\theta(H) \left( \frac{1 - \Gamma_\theta(H-1)}{\gamma_\theta(H)} (\log p(s_{H+1} \mid s_H, a_H) - \log q_\theta(s_{H+1} \mid s_H, a_H)) + H \log \gamma - \log \gamma_\theta(H) + \log r(s_H, a_H) \right) \right].$$

For *(a)*, we applied Lemma B.2 and multiplied the integrand by $\frac{q_\theta(\tau \mid H=H)\gamma_\theta(H)}{q_\theta(\tau \mid H=H)\gamma_\theta(H)} = 1$. For *(b)*, we applied Jensen's inequality. For *(c)*, we factored $p(\tau, H) = p(\tau, H)p(H)$ and $q_\theta(\tau, H) = q(\tau \mid H)\gamma_\theta(H)$. Note that under the joint distribution $p(\tau, H)$, the horizon $H \sim p(H) = \text{GEOM}(1 - \gamma)$ is independent of the trajectory, $\tau$. For *(d)*, we rewrote the expectation as an expectation over a single infinite-length trajectory and simplified the summand. For *(e)*, we recall the definition $q_\theta(\tau) = q_\theta(\tau = H = \infty)$. For *(f)*, we rewrote the integral as an expectation and wrote out the definition of the geometric distribution, $p(H)$. For *(g)*, we regrouped the difference of dynamics terms. For *(h)*, we noted used the fact that $\sum_{t=0}^{H-1} \gamma_\theta(t) + \gamma_{t=H}^\infty \gamma_\theta(t) = 1$. For *(i)*, we substituted the definition of the CDF function. For *(j)*, we rearranged terms so that all were multiplied by the discount $\gamma_\theta(H)$. Thus, we have obtained the desired result. We now prove Lemma A.1, showing that Eq. 10 becomes tight at optimality.

*Proof.*

$$\mathcal{L}_\gamma(\theta) \overset{(a)}{=} \iint q_\theta(\tau, H) (\log p(\tau, H) - \log q_\theta(\tau, H) + \log r(s_H, a_H) d\tau) dH - \log(1 - \gamma)$$

$$\overset{(b)}{=} \iint q_\theta(\tau)\gamma_\theta(H \mid \tau) (\log p(\tau) + \log p(H) - \log q_\theta(\tau) - \log \gamma_\theta(H \mid \tau) + \log r(s_H, a_H) d\tau) dH - \log(1 - \gamma) \tag{14}$$

For *(a)*, we undo some of the simplifications above, going back to Eq. 13 For *(b)*, we factor $q_\theta(\tau, H) = q_\theta(\tau)\gamma_\theta(H \mid \tau)$ and $p(\tau, H) = p(\tau)p(H)$. At this point, we can solve analytically for the optimal discount distribution, $\gamma_\theta(H \mid \tau)$:

$$\gamma_\theta^*(H \mid \tau) = \frac{p(H)r(s_H, a_H)}{\sum_{H'=0}^\infty p(H')r(s_{H'}, a_{H'})} = \frac{p(H)r(s_H, a_H)}{(1 - \gamma)R(\tau)} \tag{15}$$

In the second equality, we substitute the definition of $R(\tau)$. We then substitute Eq. 15 into our expression for $\mathcal{L}_\gamma(\theta)$ and simplify the resulting expression.

$$\mathcal{L}_\gamma(\theta) = \iint q_\theta(\tau)\gamma_\theta(H \mid \tau)\left(\log p(\tau) + \cancel{\log p(H)} - \log q_\theta(\tau) - \cancel{\log p(H)} - \cancel{\log r(s_H, a_H)} + \cancel{\log(1-\gamma)} + \log R(\tau) + \cancel{\log r(s_H, a_H)}d\tau\right)dH - \cancel{\log(1-\gamma)}$$

$$= \iint q_\theta(\tau)\gamma_\theta(H \mid \tau)\left(\log p(\tau) - \log q_\theta(\tau) + \log R(\tau)d\tau\right)dH$$

$$= \int q_\theta(\tau)\left(\log p(\tau) - \log q_\theta(\tau) + \log R(\tau)\right)d\tau.$$

In the final line we have removed the integral over $H$ because none of the integrands depend on $H$. At this point, we can solve analytically for the optimal trajectory distribution, $q_\theta(\tau)$:

$$q^*(\tau) = \frac{p(\tau)R(\tau)}{\int p(\tau')R(\tau')d\tau'}. \tag{16}$$

We then substitute Eq. 16 into our expression for $\mathcal{L}_\gamma(\theta)$, and simplify the resulting expression:

$$\mathcal{L}_\gamma(\theta) = \int q_\theta(\tau)\left(\cancel{\log p(\tau)} - \cancel{\log p(\tau)} - \cancel{\log R(\tau)} + \log\int p(\tau')R(\tau')d\tau' + \cancel{\log R(\tau)}\right)d\tau$$

$$= \log\int p(\tau)R(\tau)d\tau = \log\mathbb{E}_\pi\left[\sum_{t=0}^\infty \gamma^t r(s_t, a_t)\right].$$

We have thus shown that the lower bound $\mathcal{L}_\gamma$ becomes tight when we use the optimal distribution over trajectories $q_\theta(\tau)$ and optimal learned discount $\gamma_\theta(H \mid \tau)$. $\qquad\square$

### B.5 A lower bound for goal-reaching tasks.

Many RL problems can be better formulated as goal-reaching problems, a formulation that does not require defining a reward function. We now introduce a variant of our method for goal-reaching tasks. Using $\rho^\pi(s_{t+})$ to denote the discounted state occupancy measure of policy $\pi$, we define the goal-reaching objective as maximizing the probability density of reaching a desired goal $s_g$:

$$\max_\theta \log\rho^{\pi_\theta}(s_{t+} = s_g). \tag{17}$$

We refer the reader to Eysenbach et al. [16] for a more detailed discussion of this objective. For simplicity, we assume that the goal is fixed, noting that the multi-task setting can be handled by conditioning the policy on the commanded goal. Similar to Theorem 3.1, we can construct a lower bound on the goal-conditioned RL problem:

**Lemma B.3.** *Let initial state distribution $p_1(s_1)$, real dynamics $p(s_{t+1} \mid s_t, a_t)$, reward function $r(s_t, a_t) > 0$, discount factor $\gamma \in (0,1)$, and goal $g$ be given. Then the following bound holds for any dynamics $q(s_{t+1} \mid s_t, a_t)$ and policy $\pi(a_t \mid s_t)$:*

$$\log p^{\pi_\theta}(s_{t+} = s_g) \geq \mathbb{E}_{q^{\pi_\theta}(\tau)}\left[\sum_{t=0}^\infty \gamma^t\tilde{r}(s_t, a_t)\right], \tag{18}$$

*where*

$$\tilde{r}_g(s_t, a_t, s_{t+1}) \triangleq (1-\gamma)(\log p(s_{t+1} = s_g \mid s_t, a_t) - \log q(s_{t+1} = s_g \mid s_t, a_t) - \log(1-\gamma))$$
$$+ \log p(s_{t+1} \mid s_t, a_t) - \log q(s_{t+1} \mid s_t, a_t).$$

The proof, presented below, is similar to the proof of Theorem 3.1. The first term in the reward function, the log ratio of reaching the commanded goal one time step in the future, is similar to prior work [39]. The correction term $\log p - \log q$ incentivizes the policy to avoid transitions where the model is inaccurate, and can be estimated using a separate classifier. One important aspect of this goal-reaching problem is that it is entirely data-driven, avoiding the need for any manually-designed reward functions.

*Proof.*

$$\log \rho^{\pi_\theta}(s_{t+} = s_g) = \log \iint p(H)p(\tau \mid H = H)p(s_g \mid s_H, a_H)d\tau dH$$

$$= \log \iint p(H)\frac{p(\tau \mid H = H)}{q_\theta(\tau \mid H = H)}q_\theta(\tau \mid H = H)\frac{p(s_g \mid s_H, a_H)}{q_\theta(s_g \mid s_H, a_H)}q_\theta(s_g \mid s_H, a_H)d\tau dH - \log(1 - \gamma)$$

$$\geq \iint p(H)q_\theta(\tau \mid H = H)\left(\log p(\tau \mid H = H) - \log q_\theta(\tau \mid H = H) + \log p(s_g \mid s_H, a_H) - \log q_\theta(s_g \mid s_H, a_H)\right)d\tau dH - \log(1 - \gamma)$$

$$= \iint p(H)q_\theta(\tau \mid H = \infty)\left(\sum_{t=0}^{H}\log p(s_{t+1} \mid s_t, a_t) - \log q_\theta(s_{t+1} \mid s_t, a_t)\right) + \log p(s_g \mid s_H, a_H) - \log q_\theta(s_g \mid s_H, a_H)d\tau dH - \log(1 - \gamma)$$

$$= \int q_\theta(\tau)\int p(H)\left(\sum_{t=0}^{H}\log p(s_{t+1} \mid s_t, a_t) - \log q_\theta(s_{t+1} \mid s_t, a_t)\right) + \log p(s_g \mid s_H, a_H) - \log q_\theta(s_g \mid s_H, a_H)dHd\tau - \log(1 - \gamma)$$

$$= \int q_\theta(\tau)\sum_{t=0}^{\infty}\gamma^t\left(\log p(s_{t+1} \mid s_t, a_t) - \log q_\theta(s_{t+1} \mid s_t, a_t) + (1 - \gamma)(\log p(s_g \mid s_t, a_t) - \log q_\theta(s_g \mid s_t, a_t))\right)d\tau - \log(1 - \gamma))$$

$$= \mathbb{E}_{q_\theta(\tau)}\left[\sum_{t=0}^{\infty}\gamma^t\left(\log p(s_{t+1} \mid s_t, a_t) - \log q_\theta(s_{t+1} \mid s_t, a_t) + (1 - \gamma)(\log p(s_g \mid s_t, a_t) - \log q_\theta(s_g \mid s_t, a_t) - \log(1 - \gamma))\right)\right].$$

$\square$

Similar to the more complex lower bound presented in Eq. 10, this lower bound on goal-reaching can be modified (by learning a discount factor) to become a tight lower bound. The resulting objective would resemble a model-based version of the algorithm from Rudner et al. [39].

### B.6 Derivation of Model Objective (Eq. 9)

Our lower bound depends on entirely trajectories sampled from the learned dynamics. In this section, we show how the same objective can be expressed as an expectation of transitions. This expression is easier to optimize, as it does not require backpropagating gradients through time. We start by writing our lower bound, conditioned on a current state $s_t$.

$$\mathbb{E}_{\substack{\pi(a_t|s_t), \\ q_\theta(s_{t+1}|s_t, a_t)}}\left[\sum_{t'=t}^{\infty}\gamma^{t'-t}\tilde{r}(s_{t'}, a_{t'}) \mid s_t\right]$$

$$= \mathbb{E}_{\substack{\pi(a_t|s_t), \\ q_\theta(s_{t+1}|s_t, a_t)}}\left[\tilde{r}(s_t, a_t, s_{t+1}) + \gamma V(s_{t+1}) \mid s_t\right]$$

$$\overset{(a)}{=} \mathbb{E}_{\substack{\pi(a_t|s_t), \\ q_\theta(s_{t+1}|s_t, a_t)}}\left[(1 - \gamma)\log r(s_t, a_t) + \log\frac{C_\phi(s_t, a_t, s_{t+1})}{1 - C_\phi(s_t, a_t, s_{t+1})} - (1 - \gamma)\log(1 - \gamma) + \gamma V(s_{t+1}) \mid s_t\right]$$

In *(a)*, we substituted the definition of the augmented return. For the purpose of optimizing the dynamics model, we can ignore all terms that do not depend on $s_{t+1}$. Removing these terms, we arrive at our model training objective (Eq. 9)

## C Additional Experiments

**Fig. 6.** We compare MnM to a number of alternative model learning methods. MBPO [24] uses a standard maximum likelihood model. We implement a version of VAML [17], which augments the maximum likelihood loss with an additional temporal difference loss; the model should predict next states that have low Bellman error. Finally, we compare to a variant of the MBPO maximum likelihood model that weights transitions based on the Q values, an idea discussed (but not actually implemented) in Lambert et al. [29]. We implement this value weighting method by computing the Q values for the current states and computing a softmax over the batch dimension to obtain per-example weights.

**Fig. 8** In this experiment, we show the model MSE throughout the training of MnM-approx, similar to Fig. 7b. While MnM-approx does not optimize for MSE, the MSE is nonetheless a rough barometer for whether the model is training stably. On the DClawScrewFixed-v0 task, we observe that $\frac{3}{3}$ seeds

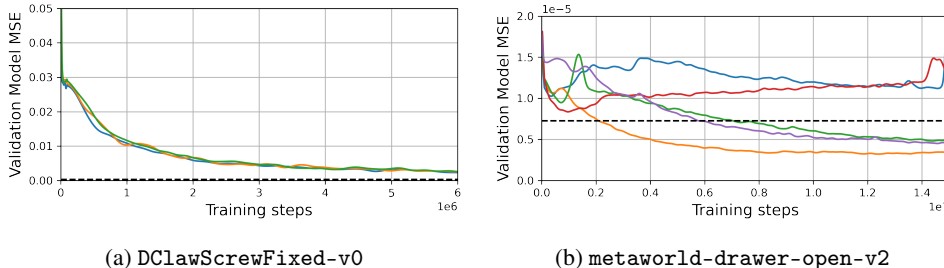

(a) `DClawScrewFixed-v0`  (b) `metaworld-drawer-open-v2`

Figure 8: **MnM-approx Model MSE**. Different lines show different random seeds, while the dashed horizontal line shows the minimum MSE of a maximum likelihood model (averaged across seeds).

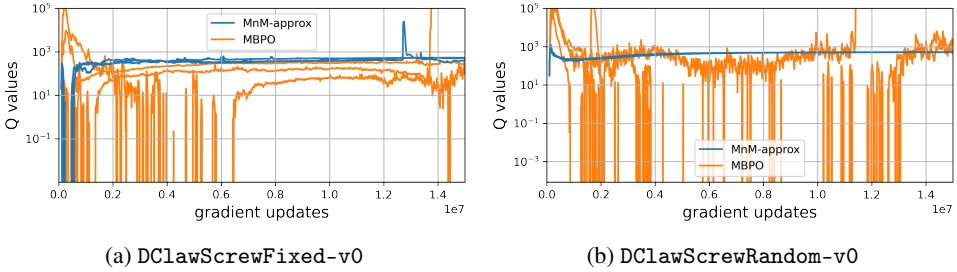

(a) `DClawScrewFixed-v0`  (b) `DClawScrewRandom-v0`

Figure 9: **MnM-approx Q-values**

all train stably, converges towards (though not quite reaching) the MSE of the maximum likelihood model. On the `metaworld-drawer-open-v2` task, the results are a bit more complicated, with $\frac{3}{5}$ seeds achieving a model MSE much lower than the maximum likelihood model, while two seeds seem to have failed to converge. Overall, these plots indicate that MnM-approx often trains stably, but some seeds can fail to converge on some environments.

**Fig. 9** In this experiment, we show the Q-values throughout the training of MnM-approx, similar to Fig. 7a. In both environments, we observe that the Q-values from MnM-approx are more stable than the Q-values from MBPO. Note how the Q-values from MBPO tend to peak early during training, suggesting that they have overestimated the true returns and then correct towards a less inflated estimate of the agent's expected return

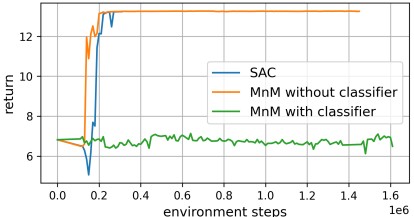

**Fig. 10.** This is a plot from a preliminary version of MnM, when we were testing the effect of the classifier term. In experiments like these, we found that the classifier term significantly hurt performance, motivating us to not include the classifier term in the "MnM-Approx" method used in the continuous control experiments.

Figure 10: Adding the classifier term to MnM degrades performance on the `metaworld-drawer-open-v2` task.

**Fig. 11** In this experiment, we ablate the two factors of the MnM augmented reward (Eq. 3, using the stochastic gridworld from Fig. 2a. The first ablation removes the logarithm, while the second ablation removes the classifier. The results, shown in Fig. 11, indicate that the logarithm term is crucial for getting good performance, but that removing classifier term has a relative small effect, and may even boost performance by a small margin. Not to read into these results too much, the aliasing experiment in Fig. 2b has already demonstrated that the classifier term is critical for ensuring good performance in the presence of function approximation.

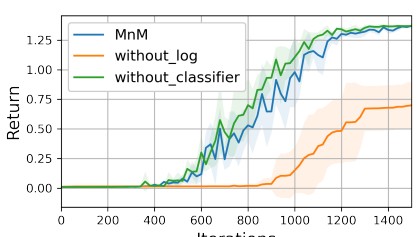

Figure 11: Stochastic gridworld ablation.

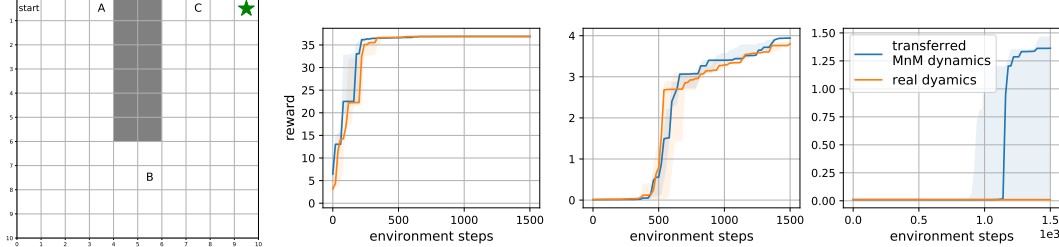

Figure 13: **Transferring the learned dynamics to a new task.** We applied Q-learning with either the true environment dynamics or with the dynamics learned by MnM on the original task (reaching the green star). The shaded region corresponds to the $[25\%, 75\%]$ region across 5 random seeds.

**Fig. 12**  In our experiments, we used very short model rollouts, only collecting a single transition from the model. Fig. 12 shows an ablation experiment on the `metaworld-drawer-open-v2` task where we increased the model horizon, finding that it uniformly degrades performance.

**Fig. 13**  In this experiment, we examine the transferability of MnM's learned dynamics model to new tasks. When learning a similar task, we might expect that the optimistic dynamics produced by MnM would be more useful than the true environment dynamics. However, when learning dissimilar tasks, the optimism for one task might hinder exploration towards solving the new task, and might be worse than using the true environment dynamics.

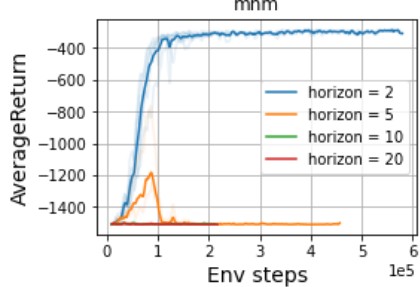

Figure 12: **Model horizon**

We tested these hypotheses using the stochastic gridworld from Fig. 2a. We took the dynamics learned by MnM (Fig. 2a (right)) and used it to solve new tasks, defined by placing the goal in different locations. As shown in Fig. 13 (right), tasks B and C are similar to the original task, in that they involve navigating around the wall; task A is not as similar to the original task, but is much easier.

To study the effectiveness of transferring the dynamics, we applied Q-learning to each of these tasks, either using the (incorrect) optimistic dynamics learned from the original task, or by using the true environment dynamics. We show the results in Fig. 13. On Tasks A and B, we observe little difference between Q-learning with the transferred dynamics model versus the true dynamics model. However, on task C, only by using the transferred dynamics is Q-learning able to solve this challenging task. In summary, these results suggest that the bias of the learned dynamics does not seem to hurt when learning dissimilar tasks, but can accelerate learning of challenging, similar tasks.

## D   Implementation Details

All experiments were run on at least three random seeds. Although we were limited by computational constraints, we find that most of our conclusions hold with $p < 0.05$, as noted in the main text.

### D.1   Algorithms

**Value Iteration (Fig. 2a _(right)_, Fig. 2b, Fig. 3b)**  For the tabular experiments that perform value iteration, we perform Polyak averaging of the policy and learned dynamics model with parameter $\tau = 0.5$. We found that the value iteration version of MnM diverged without this Polyak averaging step. Experiments were stopped when the MnM dynamics model (which depends on the value function) changed by less than 1e-6 across iterations, as measured using an $L_0$ norm. For VMBPO we used $\eta = 1$.

**Q-learning (Fig. 2a)**  For the experiments with Q-learning (both with and without the MnM components), we performed $\epsilon$-greedy exploration with $\epsilon = 0.5$. We used a learning rate of $1e - 2$. For this task alone, we compute the MnM dynamics analytically by combining the true environment dynamics with the learned value function, allowing for clearer theoretical analysis. For fair comparison, all

Table 2: **Gradient updates per real environment step**: This parameter was separately tuned for each method and each environment.

|  | SAC | MBPO | MnM |
|---|---|---|---|
| HalfCheetah-v2 | - | 40 | 20 |
| Hopper-v2 | - | 40 | 20 |
| Walker2d-v2 | - | 40 | 20 |
| metaworld-drawer-open-v2 | 40 | 40 | 40 |
| DClawPoseRandom-v0 | 20 | 20 | 20 |
| DClawTurnRandom-v0 | 40 | 40 | 40 |
| DClawScrewFixed-v0 | 40 | 40 | 40 |
| DClawScrewRandom-v0 | 40 | 40 | 40 |

methods receive the same amount of data, perform the same number of updates, and are evaluated using the real environment dynamics. For VMBPO we used $\eta = 1$.

**SAC for continuous control tasks.** We used the SAC implementation from TF-Agents [21] with the default hyperparameters.

**MBPO for continuous control tasks.** We implement MBPO on top of the SAC implementation from TF-Agents [21]. Unless otherwise mentioned, we take the default parameters from this implementation. We use an ensemble of 5 dynamics models, each with 4 hidden layers of size 256. The dynamics model predicts the whitened difference between the next state and the current state. That is, to obtain the prediction for the next state, the predictions are scaled by a per-coordinate variance, shifted by a per-coordinate mean, and then added to the current state. These whitened predictions are clipped to have a minimum standard deviation of 1e-5; without this, we found that the MBPO model resulted in numerical instability. The model is trained using the standard maximum likelihood objective, with all members of the ensemble being trained on the same data. To sample data from this model we perform 1-step rollouts, starting at states visited in the true dynamics. We perform one batch of rollouts in parallel using a batch size of 256. To sample the corresponding action, with probability 50% we take the action that was executed in the true dynamics; with probability 50% we sample an action from the current policy. We found that this modification slightly improves the results of MBPO. We use a batch size of 256. We have two replay buffers: the model replay buffer has size 256e3 and the replay buffer of real experience has size 1e6. At the start of training, we collect 1e4 transitions from the real environment, train the dynamics model on this experience for 1e5 batches, and only then start training the policy. We use a learning rate of 3e-4 for all components. To stabilize learning, we maintain a target dynamics model using an exponential moving average ($\tau = 0.001$), and use this target dynamics model to sample transitions for training. We update the model, policy, and value functions at the same rate we sample experience from the learned model, which is more frequently than we collect experience from the real environment (see Table 2).

**MnM for Continuous Control Tasks** We implement MnM on top of the SAC implementation from TF-Agents [21]. Unless otherwise mentioned, we take the default parameters from this implementation. Our model architecture is exactly the same as our MBPO implementation, and we follow the same training protocol.

Unlike MBPO, MnM also learns a classifier for distinguishing real versus model transitions. The classifier architecture is a 2 layers neural network with 1024 hidden units in each layer. We found that this large capacity was important for stable learning. We add input noise with $\sigma = 0.1$ while training the classifier. We whiten the inputs to the classifier by subtracting a coordinate-wise mean and dividing by a coordinate-wise standard deviation. When training the classifier, we take samples from both the dynamics model and the target dynamics model as negative examples, finding that this stabilizes learning somewhat. Following the suggestion of prior work [40], we use one-sided label smoothing with value 0.1, only smoothing the negative predictions and not the positive predictions.

We found that gradient penalties and spectral normalization decreased performance. We found that automatically tuning the classifier input noise also decreased performance. We found that mixup had little effect. We found that the loss would often plateau around 1e4 batches, but would eventually start decreasing again after 2e4 - 2e5 batches.

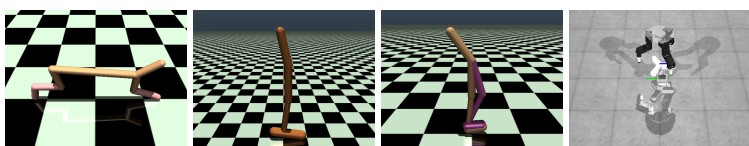

Figure 14: **Environments**: Our experiments included three tasks from OpenAI Gym and four tasks from ROBEL.

Like the MBPO model, we first collect 1e4 transitions of experience from the real environment using a random policy, then train the dynamics model and classifier for 1e5 batches, and only then start updating the policy. Because the Q values are poor at the start of training, we only add the value term to the model loss (resulting in the optimistic dynamics model) after 2e5 batches (1e5 batches of model training, then 1e5 batches of model+policy training). To further improve stability, we compute the value term in the model loss by taking the minimum over two target value functions (like TD3 [18]). We update the model, classifier policy, and value functions at the same rate we sample experience from the learned model, which is more frequently than we collect experience from the real environment (see Table 2).

### D.2 Environments

This section provides details for the environments used in our experiments. We visualize the environments in Fig. 14.

**Gridworld for Fig. 2a.** This task is a $10 \times 10$ gridworld with obstacles shown in Fig. 2a. There are four discrete actions, corresponding to moving to the four adjacent cells. With probability 50%, the agent's action is ignored and a random action is taken instead. The agent starts in the top-left cell. The agent receives a reward of +0.001 at each time step and a reward of +10 when at the goal state. Episodes have 200 steps and we use a discount $\gamma = 0.9$.

**Gridworld for Fig. 2b.** This task is a $15 \times 15$ gridworld with obstacles shown in Fig. 2b. There are four discrete actions, corresponding to moving to the four adjacent cells. The dynamics are deterministic. The agent starts in the top-left cell. The reward is +1.0 at every state except the goal state, where the agent receives a reward of +100.0. We use $\gamma = 0.9$ and compute optimal policies analytically using value iteration. We implement the aliasing by averaging together the dynamics for each block of $3 \times 3$ states. Importantly, the averaging was done to the *relative* dynamics (e.g., action 1 corresponds to move right) not the *absolute* dynamics (e.g., action 1 corresponds to moving to state (3, 4)). To handle edge effects, we modified the averaged dynamics so that the agent could not exit the gridworld. We computed the classifier analytically using the true and learned dynamics models. However, the augmented rewards become infinite because the learned model assigns non-zero probability to transitions that cannot occur under the true dynamics. This is not a failure of our theory, as the optimal policy would choose to never visit these states, but it presents a challenge for optimization. We therefore added label smoothing with parameter 0.7 to the classifier.

**Gridworld for Fig. 2a** This task is a $10 \times 10$ gridworld with obstacles shown in Fig. 2a *(Right)*. There are four discrete actions, corresponding to moving to the four adjacent cells. With probability 90%, the agent's action is ignored and a random action is taken instead. The agent starts in the top-left cell. The rewards depend on the Manhattan distance to the goal: transitions that lead away from the goal have a reward of +0.001, transitions that do not change the distance to the goal have a reward of +1.001, and transitions that decrease the distance to the goal have a reward of +2.001. We use $\gamma = 0.5$ and compute values and returns analytically using value iteration.

**Gridworld for Fig. 3b.** This task used the same dynamics as Fig. 2a. The one change is that the agent receives a reward of +1.0 at each time step and a reward of +10.0 at the goal state. We estimate the more complex lower bound (Eq. 10) by also learning the discount factor. However, since we currently do not have a method for learning non-Markovian dynamics to fully optimize this lower bound, we do not expect the lower bound to become tight.

`HalfCheetah-v2, Hopper-v2, Walker2d-v2.` These tasks are taken directly from the OpenAI benchmark [7] without modification.

`metaworld-drawer-open-v2.` This task is based on the `drawer-open-v2` task from the Metaworld benchmark [50]. To increase the difficulty of this task, we remove the reward shaping term (`reward_for_caging`) and just optimize the reward for opening the drawer (`reward_for_opening`).

`DClawPoseRandom-v0, DClawTurnRandom-v0, DClawScrewFixed-v0, DClawScrewRandom-v0.` These tasks are taken directly from the ROBEL benchmark [1] without modification.

### D.3 Differences between MnM and MnM-Approx

Both MnM and MnM-Approx learn the same dynamics classifier, and both use the same GAN-like objective (Eq. 9) to update the model. Both perform the same policy updates. The difference is the reward function used to update the Q-function:

$$\tilde{r}(s_t, a_t, s_{t+1}) = (1 - \gamma) \log r(s_t, a_t) + \log \left( \frac{p(s_{t+1} \mid s_t, a_t)}{q(s_{t+1} \mid s_t, a_t)} \right) \qquad \text{(MnM)}$$

$$\tilde{r}(s_t, a_t, s_{t+1}) = r(s_t, a_t). \qquad \text{(MnM-Approx)}$$