# OpenReview forum: "Mismatched No More: Joint Model-Policy Optimization for Model-Based RL"
_NeurIPS.cc/2022/Conference — NeurIPS 2022 Accept_

### Official Review · Reviewer_27GT · 2022-07-10

**Rating:** 8
**Confidence:** 5
**Soundness:** 3 good
**Presentation:** 3 good
**Contribution:** 4 excellent

**Summary:**

This paper aims to solve objective mismatch in model-based RL which claims that the objective of the model learning is different from the objective of policy learning. The authors propose Mismatch no More (MnM) which optimizes the lower bound of the expected return to jointly training the dynamics model and the policy. To optimize this lower bound practically, they introduce a modified reward function which is estimated by a trained GAN-like classifier. Experimental results show that MnM can achieve performance competitive with prior model-based methods, and better performance on certain hard exploration tasks.

**Questions:**

See Weaknesses above.

**Limitations:**

See Weaknesses above.

**Strengths And Weaknesses:**

Strengths:

1. To my knowledge, the idea is novel, and the idea of training a classifier to distinguish the real transitions from the fake transitions is very interesting.
2. The paper is well-written and easy to follow.

Weaknesses:

1. My main concern is about the experiments. This paper aims to solve the objective mismatch in MBRL and it is very related to Value-Awareness Model-based RL. However, they don't compare with the SOTA Value-Awareness Model-based RL methods such as VaGarm [1]  in both OpenAI Gym and ROBEL evaluations. In ROBEL environment, they even don't compare with VMBPO.

2. Besides, an important part of dyna-style model-based RL is that the learned dynamics model can perform multi-step rollout for policy learning to improve sample efficiency. I want to know whether the learned dynamics model of MnM can perform multi-step rollout, because this model is completely dependent on the value function, and has not learned the knowledge related to transition. How to control model compounding error in MnM if multi-step rollout is possible?

Typos:

In Eq.8, does $\tilde{Q}_\psi$ mean ${Q}_\psi$? I didn't find the definition of $\tilde{Q}_\psi$.

[1] Voelcker, Claas A., et al. "Value Gradient weighted Model-Based Reinforcement Learning." International Conference on Learning Representations. 2022.

---

> ### Author Response · Authors · 2022-08-02
> **Response to 27GT**
>
> We thank the reviewer for the time they put into their review, and for their suggestions for improving the work. The reviewer's two main concerns seem to be about lacking a comparison to recent value-aware methods and and understanding of how the model horizon effects MnM. We have attempted to address these concerns by adding the VaGraM baseline in the Mujoco results experiment (Figure 4) and including an ablation experiment studying the effect of model horizon (new Figure 12). Please let us know if these changes and responses address all of the concerns in the review, or if there are remaining concerns that we can address.
>
> > This paper … is very related to Value-Awareness Model-based RL. However, they don't compare with the SOTA Value-Awareness Model-based RL methods such as VaGarm.
>
> We ran an additional experiment to compare VaGraM to MnM-approx and the other baselines, using the official implementation [7]. The initial results, shown in Figure 4, indicate that VaGraM performs worse than MnM-approx.
>
> > In ROBEL environment, they even don't compare with VMBPO.
>
> VMBPO did not release code, so we were only able to compare on the environments where the authors provided learning curves (personal correspondence).
>
> > How to control model compounding error in MnM if multi-step rollout is possible?
>
> As suggested by the reviewer, we ran an ablation experiment on the metaworld-drawer-open-v2 task studying the effect of model horizon. The results, shown in Appendix Figure 12 show that MnM-approx does not learn when using rollouts of 5, 10, or 20. The problem of learning from longer rollouts remains a hard and open problem [2], one that is orthogonal to the aims of this paper.
>
> > In Eq.8, does $\tilde{Q}_\psi$ mean $Q_\psi$?
>
> Yes. We have fixed this typo.
>
>
> [1] https://github.com/pairlab/vagram

---

> ### Author Response · Authors · 2022-08-05
> **Have the revisions addressed the reviewer's concerns?**
>
> Dear Reviewer,
>
> Thank you for raising a number of questions and concerns in the initial review. **Have the new experiments, revisions and responses below addressed all the concerns?** We would be happy to address any additional questions or concerns.

---

> > ### Author Response · Authors · 2022-08-07
> > **Additional concerns?**
> >
> > Dear Reviewer,
> >
> > We hope that the new experiments and revisions have addressed the concerns raised in the review. **Does the reviewer have any additional concerns, or suggestions for further improving the paper?** We appreciate the feedback, as it helps to make the paper stronger.

---

> > ### Comment · Reviewer_27GT · 2022-08-08
> > **Response**
> >
> > Thanks for your hard working! Most of my concerns have been addressed.
> > I wonder why the performance of VaGram is significantly lower than the original paper.
> > I will keep my initial score.
> >
> > Best

---

> > > ### Author Response · Authors · 2022-08-08
> > > **Author response**
> > >
> > > Dear Reviewer,
> > >
> > > Thanks for getting back to us, and for confirming that the concerns have been addressed. Of course, if any additional concerns do arise, please let us know and we'll do our best to run additional experiments or revise the paper to address these.
> > >
> > > >  I wonder why the performance of VaGram is significantly lower than the original paper.
> > >
> > > To confirm, the question is why the VaGram curves that we have added are worse than those reported in the VaGram paper? One hypothesis is that the official VaGram repository is not exactly the same as the code used to produce the plots in the VaGram paper. We reached out to the authors of the VaGram paper, and they said that the repository should reproduce the plots in the paper, but there might be a bug in the code, and that bug might cause some of the figures to be "slightly off."
> > >
> > > **More details**: We believe this is a fair comparison because use directly used the official GitHub repository [1] and chose the hyperparameters (`num_layers`, `hid_size`, and `model_batch_size`) to exactly match those used in our paper. Precisely, we ran commands like the following:
> > >
> > > ```python3 -m mbrl.examples.main seed=0 algorithm=mbpo overrides=mbpo_hopper dynamics_model.model.num_layers=4 dynamics_model.model.hid_size=256 overrides.model_batch_size=256```
> > >
> > > These hyperparameters are within the range considered in the VaGram paper, so we think it is unlikely that these were simply bad hyperparameters for VaGram.
> > >
> > >
> > > [1] https://github.com/pairlab/vagram

---

> > > > ### Comment · Reviewer_27GT · 2022-08-08
> > > > **Thank you for your response**
> > > >
> > > > I have read their code and there is actually a bug in their implementation. All my questions are addressed.
> > > > Since this paper is really novel and interesting, I will raise my score from 6 to 8.

---

> > > > > ### Author Response · Authors · 2022-08-09
> > > > > **Thanks!**
> > > > >
> > > > > Thanks for the update! Please do let us know if there are any additional questions or concerns!

---

### Official Review · Reviewer_FWhQ · 2022-07-10

**Rating:** 7
**Confidence:** 3
**Soundness:** 3 good
**Presentation:** 3 good
**Contribution:** 3 good

**Summary:**

The proposed method provides a strategy based on a standard variational inference framework to perform model-based reinforcement learning by addressing the objective-mismatch issue. The algorithm “Mismatched-no-More” (MnM) jointly optimizes the model and policy with the help of an objective that serves as a “global” lower bound for the expected return. Practically, the approach uses a new augmented reward function containing the error between learned and real transition probabilities simplified with the help of a GAN-like classifier to differentiate between the two. The efficacy of the method is validated on 2 tabular MDPs, 3 locomotion tasks, and 4 manipulation tasks.

**Questions:**

- Given that “MnM-approx” is need to ensure appropriate exploration, the reviewer is curious to see the results of using MnM-approx in the tabular settings as well. Is training the policy and model with the same objective really needed at all in practice?
- I appreciate the Manipulation experiments but since the most similar algorithm to MnM is VMBPO, the authors should at least conduct experiments in all the environments shown by the original VMBPO paper. Also, the authors should properly justify the discrepancy between the VMBPO scores shown in the submitted paper and the original one.
- Is it possible to include VMBPO in Fig. 5 since the authors have included it for the tabular settings? That would complete it.
- By looking at the HalfCheetah-v2 model prediction results, the reviewer lacks clarity in realizing the advantage of using such a model. While taking an action in the initial state topples the robot in the real world, it helps the robot to move forward in the learned model. Can the authors bring some clarity given that the final evaluation of the policy is on the real environment?
- It is confusing to see the performance analysis on `DClawScrewFixed-v0`, MSE analysis on `DClawScrewRandom-v0` task and Q-value analysis on `metaworld-drawer-open-v2`. Since, the reviewer can still see a noticeable difference in the performance between MnM-approx and MBPO on `DClawScrewFixed-v0`, something more convincing will be to conduct all the analysis on the same. Moreover, adding the performance of the additional environments as the authors already have it will be the most appreciated.


**Limitations:**

The reviewer acknowledges all the limitations of the approach as mentioned by the authors and also agrees with the same. However, due to a very selective analysis of the comparison performance, the experimental segment is still not convincing enough.

**Strengths And Weaknesses:**

The paper is well written with several strong contributions:

- The paper clearly distinguishes itself from very similar prior works (like VMBPO) in terms of its theoretical contribution to maximizing the lower bound on expected return. To the best of the reviewer’s knowledge, the approach of using a GAN-like objective for model learning is novel.

- The additional analysis of model performance, model exploitation, and training performance of the MnM model with the MSE objective adds additional support to the claims made in the paper.

However, the difference between the proposed theoretical formulation and the experimental implementation makes it difficult to realize the true significance of using MnM. The approach is proposed with the intention of optimizing the same objective for policy and model learning but conducts most of the experiments only with model learning.

- Further, the comparison results are not clear on what advantage we get from MnM. In continuous control experiments, the authors show a clear analysis of the superiority of MnM-approx over other methods only on `DClawScrewRandom-v0` out of the seven environments.
- On all the OpenAI Gym environments, the improvement in performance over MBPO is not considerable, given that an additional classifier was trained to replace the maximum likelihood objective.
- Finally, the reviewer could not correspond the VMBPO scores presented in the paper with their published one. Looking at the original values, there seems to be hardly any difference between the performance of VMBPO and MnM which is expected given that they are solving the same objective eventually. (referring: https://www.ijcai.org/proceedings/2021/0316.pdf)

---

> ### Author Response · Authors · 2022-08-02
> **Response to FWhQ**
>
> We thank the reviewer for the time they put into their review, and for their suggestions for improving the work. We believe the reviewer's main concern is about the approximate version of MnM used in the continuous control experiments. We agree that the gains relative to the baselines are small on most tasks, and will revise the paper to clarify this point. Nonetheless, we believe that the paper has both conceptual value (in relating GANs to model-based RL) and theoretical contributions (a new lower bound for model-based RL). To address reviewer concerns, we have added 2 new figures to the paper and revised the paper to incorporate the feedback. Please let us know if these changes and responses address all of the concerns in the review, or if there are remaining concerns that we can address.
>
> > Further, the comparison results are not clear on what advantage we get from MnM.
>
> We believe that, despite not achieving state-of-the-art results across every benchmark, the method still has a lot of value for the community due to the unified objective and conceptual contribution to the study of model-based RL.
>
> > the reviewer could not correspond the VMBPO scores presented in the paper with their published one
>
> We received the learning curves from VMBPO from the original authors. Looking at the arXiv version of the VMBPO paper [1], the curves look identical to the ones in our paper.
>
> > the reviewer is curious to see the results of using MnM-approx in the tabular settings as well. Is training the policy and model with the same objective really needed at all in practice?
>
> The "original rewards" (orange) line in Figure 2b is the same as MnM-approx: it trains the dynamics model as prescribed by MnM, but updates the policy using the original task rewards (not Eq. 3). This experiment illustrates the importance of training the model and policy with the same objective, or of at least training the policy to be aware of inaccuracies in the learned dynamics model. By incorporating the model in the policy objective, MnM learns a policy that navigates around the obstacle (see L280 – L290).
>
> > the authors should at least conduct experiments in all the environments shown by the original VMBPO paper
>
> We will work on doing this for the final version. We note that the current number of continuous control environments (7) compares favorably to many prior model-based RL papers. For example, the VaGraM baseline suggested by Reviewer 27GT [2, ICLR 2021] studies just two tasks in the main text (Pendulum and Hopper). Other recent model-based RL papers accepted to top-tier venues run experiments solely on minigrid [3, NeurIPS 2021] or on the three simplest Mujoco tasks, pendulum/cartpole/acrobot [4, ICML 2021].
>
> > Is it possible to include VMBPO in Fig. 5 since the authors have included it for the tabular settings?
>
> VMBPO did not release code, so we are only able to perform the comparison with VMBPO on the continuous control environments where they have released the learning curves. [However, implementing VMBPO in the tabular setting is straightforward, so we do have comparisons there.]
>
> > By looking at the HalfCheetah-v2 model prediction results, the reviewer lacks clarity in realizing the advantage of using such a model. Can the authors bring some clarity given that the final evaluation of the policy is on the real environment?
>
> Although learning in the optimistic model may seem undesirable at first, the model optimism helps the agent stay upright and continue collecting valuable experience, rather than falling and wasting the rest of the episode. We will clarify this example in the paper.
>
> > It is confusing to see the performance analysis on DClawScrewFixed-v0, MSE analysis on DClawScrewRandom-v0 task and Q-value analysis on metaworld-drawer-open-v2 … Something more convincing will be to conduct all the analysis on the same.
>
> As suggested by the reviewer, we additionally ran the MSE analysis on DClawScrewFixed-v0 and metaworld-drawer-open-v2, and additionally ran the Q-value analysis on DClawScrewFixed-v0 and DClawScrewRandom-v0. The results can be found in Appendix Figures 10 and 11. The same conclusions generally hold for these environments: the Q-values are more stable for MnM-approx than for MBPO, and the model MSE decreases (though there is instability in 2 seeds for 1 environment).
>
> **Have these responses addressed the reviewer's concerns?** Does the reviewer have additional comments or suggestions for improving the paper? We look forward to continuing the discussion.

---

> > ### Author Response · Authors · 2022-08-02
> > **Response to FWhQ**
> >
> >
> >
> > [1] https://arxiv.org/pdf/2006.05443.pdf
> >
> > [2] Voelcker, Claas A., et al. "Value Gradient weighted Model-Based Reinforcement Learning." International Conference on Learning Representations. 2021.
> >
> > [3] Zhao, Mingde, et al. "A consciousness-inspired planning agent for model-based reinforcement learning." Advances in Neural Information Processing Systems 34 (2021): 1569-1581.
> >
> > [4] Yildiz, Cagatay, Markus Heinonen, and Harri Lähdesmäki. "Continuous-time model-based reinforcement learning." International Conference on Machine Learning. PMLR, 2021.

---

> > ### Comment · Reviewer_FWhQ · 2022-08-09
> > **Rebuttal Acknowledgement Response**
> >
> > Hi authors,
> >
> > Thank you for addressing my questions.
> >
> > I agree with the proposed direction's novelty and strength and acknowledge its theoretical and conceptual value. My primary concern was the use of two different settings in the paper, one for theoretical evidence showing the learning of a unified objective and the other as a typical model-based RL with a GAN-framework-based dynamics model. I think that the change in the reward formulation from MnM to MnM-approx significantly affects the training of the policy.
> >
> > Apart from this, I still lack clarity on the advantage of model optimism. If irrespective of the actions, the model predicts an upright pose (where the real pose is the opposite), this simply misleads the policy.
> >
> > Finally, I appreciate the additional results and understand the proposed method's contributions.
> >
> > I will raise my score from 6 to 7.

---

> > > ### Author Response · Authors · 2022-08-09
> > > **Thanks for the update!**
> > >
> > > Dear Reviewer,
> > >
> > > Thank you for the response. This discussion is helpful for us to clarify the contributions of the paper, including the capabilities and limitations of the proposed method.
> > >
> > > > difference between MnM and MnM-approx
> > >
> > > We agree that this change affects the training of the policy; in response to Reviewer Y1gL, we added a new Figure 8 to the appendix showing that including the classifier term decreases performance in the continuous control setting.
> > >
> > > One explanation, alluded to to Section 4.3, is that the classifier term is anti-exploratory, in a way that can actually be made precise. If the true dynamics and learned dynamics are both Gaussian, then the classifier term (in expectation, if estimated perfectly) looks like the mean-squared error of the learned dynamics model [1]. Rewards based on model error are commonly used for exploration [2], but the classifier term looks like a _negated_ version of these exploration bonuses. This provides some intuition for why the classifier term would hurt performance in some scenarios (continuous control tasks, but not tabular settings).
> > >
> > > >  If irrespective of the actions, the model predicts an upright pose (where the real pose is the opposite), this simply misleads the policy.
> > >
> > > We agree models that ignore actions would be challenging to use for model-based RL. However, the models learned by MnM do still pay attention to the actions. If an action always leads to some outcome under the real model (e.g., if an action corresponding to "tripping" always leads to a "falling" state), then it will always lead to the same outcome under the MnM model [3].
> > >
> > > **Please let us know if any additional questions or concerns arise.**
> > >
> > > ------------------
> > >
> > > [1] Precisely, it corresponds to a KL divergence, and the KL divergence between Gaussians with fixed variances is a MSE (https://stats.stackexchange.com/a/7449).
> > >
> > > [2] Stadie, B. C., Levine, S., and Abbeel, P. (2015). Incentivizing exploration in reinforcement learning with deep predictive models. arXiv preprint arXiv:1507.00814.
> > >
> > > [3] In L159, if $p(\tau) = 0 \implies q(\tau) = 0$.

---

> ### Author Response · Authors · 2022-08-05
> **Have the revisions addressed the reviewer's concerns?**
>
> Dear Reviewer,
>
> Thank you for raising a number of questions and concerns in the initial review. **Have the new figures, revisions and responses below addressed all the concerns?** We would be happy to address any additional questions or concerns.

---

> > ### Author Response · Authors · 2022-08-07
> > **Additional concerns?**
> >
> > Dear Reviewer,
> >
> > We hope that the new figures and revisions have addressed the concerns raised in the review. **Does the reviewer have any additional concerns, or suggestions for further improving the paper?** We appreciate the feedback, as it helps to make the paper stronger.

---

### Official Review · Reviewer_Y1gL · 2022-07-10

**Rating:** 6
**Confidence:** 4
**Soundness:** 2 fair
**Presentation:** 3 good
**Contribution:** 2 fair

**Summary:**

Model-Based Reinforcement Learning (MBRL) is usually treated as a supervised learning problem where we first fit the data from previously observed data by reducing MSE and then do planning using this learned model. This paper highlights that this way of learning the model for planning leads to the objective mismatch problem: The model training objective is disconnected from the policy optimization process. While highlighting the problems caused by this objective mismatch, the authors propose a workaround by a unified objective for both model and policy learning. More specifically by viewing model learning as a latent variable problem, the author leverage theory from VI and propose a novel lower bound (similar to ELBO) which they maximize during the learning process. Then the authors propose a practical GAN-style algorithm where the aim is to learn a model which fools the discriminator by generating realistic-looking fake trajectories. The policy network is trained to maximize the log of the reward. Since all these optimizations are done with just a single objective, the model learning and policy learning phases are coherent resulting in learning high-return policies.   The authors support the theory presented in the main text from a series of experiments on simple tasks demonstrating the proposed algorithm's efficacy.

**Questions:**

1. Line 163 states “...... overestimate the policy’s return, violating lemma 3.1”. However, I skimmed through the entire paper searching for a formal statement for lemma 3.1 with little success. Could the authors state where is lemma 3.1 in the paper?
2. Regarding difficulties of including the classifier term while doing policy optimization for continuous control tasks, I see that this is just stated without any experimental evidence. This makes it unclear for readers to understand if this is indeed causing an issue and if yes, in what way it is hurting the exploration.
3. The authors say that they performed experiments on the OpenAI benchmark while citing the OpenAI gym paper. As far as I know, there are several environments in the OpenAI gym, but the authors seem to perform experiments only on 3 environments which makes it hard to comment on the scope of the algorithm. Could you please clarify a) the motivation behind picking these specific environments and b) the validity of your approach on the entire mujoco suit?
4.Since all of the continuous control experiments are based on MnM approx algorithm, it would be helpful to have pseudocode highlighting the differences between MnM and MnM approx.



**Limitations:**

The limitations are clearly mentioned in the paper and I am happy that the authors are transparent about the validity of their method. While having a single objective to train both policy and model is helpful, the authors do mention it seems to scale poorly with the problem complexity and one must employ some tricks as the authors did in MnM approx method to handle these inefficiencies. This begs the following question: Is single objective training a right way to go about building model based algorithms? If yes, why do you think so?


**Strengths And Weaknesses:**

Strenghts:
1. The contributions are clear and the authors did a good job of reiterating their contributions every now and then. This helped in getting an overall big picture
2. The writing is very clear.
3. The authors discussed the limitations of their approach clearly which is commendable.

Weaknesses:
1. Most of the experiments are toy-ish/limited and I think performing more experiments on a wide variety of tasks and benchmarks would be helpful in understanding the efficacy of MnM algorithm.
2. MnM approx algorithm is not completely clear.
3. The theory could be stated more formally (More on this in the next section)
4. No ablation studies
5. Experiments are performed on limited random seeds (3 seeds) which makes it hard to comment on the validity of the reported learning curves.

---

> ### Author Response · Authors · 2022-08-02
> **Response to Y1gL**
>
> We thank the reviewer for the time they put into their review, and for their suggestions for improving the work. We have revised the paper to include additional experiments and analysis, and attempt to address additional questions/concerns below. Please let us know if these changes and responses address all of the concerns in the review, or if there are remaining concerns that we can address.
>
> > Most of the experiments are toy-ish/limited and I think performing more experiments on a wide variety of tasks and benchmarks would be helpful in understanding the efficacy of MnM algorithm… Could you please clarify a) the motivation behind picking these specific environments and b) the validity of your approach on the entire mujoco suit?
>
> While the paper does make extensive use of tabular settings to study MnM in the absence of noise, our evaluation includes 7 continuous control tasks from 2 benchmarks, including the most widely used MuJoCo continuous control domains (HalfCheetah, Walker2d, and Hopper), and several tasks from ROBEL modeled on realistic robotic systems. As suggested by Reviewer 27GT, we have added an additional baseline to these results. We choose the three locomotion tasks because they are the most commonly used benchmarks, and choose the ROBEL tasks because we expected their more realistic contact dynamics to pose a challenge to model-based methods.
>
> While we will work on adding comparisons to the entire mujoco suite, we believe that the current comparison to 7 environments compares favorably to many prior model-based RL papers. For comparison, here is a sample of model-based RL papers accepted at top-tier conferences:
> * [1, ICLR 2021]: 2 tasks (pendulum, hopper)
> * [2, NeurIPS 2021]: ~1 task (minigrid, with varying difficulty)
> * [3, ICML 2021] 3 tasks (pendulum, cartpole, acrobot)
> * [4, NeurIPS 2020] 3 tasks (four rooms, catch, cartpole)
> * [5, NeurIPS 2020] 4 tasks (halfcheetah, hopper, walker2d, ant)
> * [6, NeurIPS 2020] 4 tasks (reacher, pusher, sparse-reacher, halfcheetah)
>
> > MnM approx algorithm is not completely clear. Pseudocode highlighting the differences between MnM and MnM approx.
>
> We have added this to Appendix D.2.
>
> > The theory could be stated more formally. Where is lemma 3.1?
>
> "Lemma 3.1" should refer to "Theorem 3.1," a typo which we have now fixed. **Does the reviewer have any additional concerns about the formality, clarity, or correctness of the theory?**
>
> > No ablation studies
>
> Figures 2 and 3 already contain some ablation studies, and we have run an additional ablation study on the stochastic gridworld and provide results in Appendix Figure 9.
>
> For example, Figure 2b shows that removing the reward augmentation suggested by our theory (Eq. 3) results in suboptimal performance. Similarly, Figure 3 shows that removing just the logarithm transformation from the reward augmentation ("VMBPO") still causes the method to achieve suboptimal performance and provides an invalid lower bound.
>
>
> > Experiments are performed on limited random seeds (3 seeds) which makes it hard to comment on the validity of the reported learning curves.
>
> The tabular experiments are performed on at least 5 random seeds, and we will add more seeds to the continuous control experiments for the final version.
>
> > Regarding difficulties of including the classifier term while doing policy optimization for continuous control tasks, I see that this is just stated without any experimental evidence.
>
> As suggested by the reviewer, we have added a figure (Appendix Figure 8) to the paper showing that adding the classifier term hurts performance. This was a preliminary experiment, so we only ran one seed, but it nonetheless shows a stark degradation in performance from adding the classifier term.
>
> > Is single objective training a right way to go about building model based algorithms? If yes, why do you think so?
>
> The overall goal of reinforcement learning is to recover a policy with the highest expected reward, and hence it makes sense that all the parts of the algorithm should be optimizing (a bound on) this overall objective. Hence, we believe that a single training objective (or approximation thereof) is indeed the right way to go.
>
> **Have these responses addressed the reviewer's concerns?** Does the reviewer have additional comments or suggestions for improving the paper? We look forward to continuing the discussion.

---

> > ### Author Response · Authors · 2022-08-02
> > **Response to Y1gL**
> >
> >
> > [1] Voelcker, Claas A., et al. "Value Gradient weighted Model-Based Reinforcement Learning." International Conference on Learning Representations. 2021.
> >
> > [2] Zhao, Mingde, et al. "A consciousness-inspired planning agent for model-based reinforcement learning." Advances in Neural Information Processing Systems 34 (2021): 1569-1581.
> >
> > [3] Yildiz, Cagatay, Markus Heinonen, and Harri Lähdesmäki. "Continuous-time model-based reinforcement learning." International Conference on Machine Learning. PMLR, 2021.
> >
> > [4] Grimm, Christopher, et al. "The value equivalence principle for model-based reinforcement learning." Advances in Neural Information Processing Systems 33 (2020): 5541-5552.
> >
> > [5] Kidambi, Rahul, et al. "Morel: Model-based offline reinforcement learning." Advances in neural information processing systems 33 (2020): 21810-21823.
> >
> > [6] Curi, Sebastian, Felix Berkenkamp, and Andreas Krause. "Efficient model-based reinforcement learning through optimistic policy search and planning." Advances in Neural Information Processing Systems 33 (2020): 14156-14170.

---

> ### Author Response · Authors · 2022-08-05
> **Have the revisions addressed the reviewer's concerns?**
>
> Dear Reviewer,
>
> Thank you for raising a number of questions and concerns in the initial review. **Have the additional experiments, revisions and responses below addressed all the concerns?** We would be happy to address any additional questions.

---

> > ### Author Response · Authors · 2022-08-07
> > **Additional concerns?**
> >
> > Dear Reviewer,
> >
> > We hope that the new experiments and revisions have addressed the concerns raised in the review. **Does the reviewer have any additional concerns, or suggestions for further improving the paper?** We appreciate the feedback, as it helps to make the paper stronger.

---

> > > ### Comment · Reviewer_Y1gL · 2022-08-07
> > > **Concerns Addressed**
> > >
> > > Hello authors,
> > >
> > > My concerns have been addressed so thank you so much. I am raising my score from 5 to 6.

---

> > > > ### Author Response · Authors · 2022-08-07
> > > > **Thanks!**
> > > >
> > > > Thanks for the update! Please do let us know if there are any additional questions or concerns!

---

### Official Review · Reviewer_HuRX · 2022-07-15

**Rating:** 8
**Confidence:** 3
**Soundness:** 3 good
**Presentation:** 3 good
**Contribution:** 4 excellent

**Summary:**

The reviewed paper is concerned with a fundamental challenge of model-based reinforcement learning, namely that the objective used to train the dynamics model may not align with the final objective of return maximization. To this end, the authors propose a globally valid lower bound on the expected return from which both a dynamics model and policy optimization objective can be derived. Intuitively, the dynamics model is trained to produce transitions which are hardly distinguishable from real transitions, resembling a GAN objective. Additionally, policy and dynamics model training are entangled, with measures of each appearing in the training objectives of the other. The authors show the functioning of their method convincingly on a set of toy-, but also complex high-dimensional manipulation environments.


**Questions:**

Would it be possible to combat the exploration problem by only modifying the reward with the classifier
term for experience which comes from the model, and use the original reward for transitions
observed on the real system? I.e., train $Q$ and $\pi$ on both type of transitions, but only
modify the reward on the simulated ones? Intuitively, this would make sense to me, as it would give lower
reward to potentially mismatching transitions which deviate from the real system.


**Limitations:**

One advantage of MBRL, in addition to often being more sample-efficient than purely model-free methods,
is the possibility to quickly transfer to different tasks (by generating transitions from the model and
training the policy with the changed reward, ideally requiring no additional interactions on the real system).
It would be beneficial if the authors could comment on if their proposed method also gives advantages
in these situations where the model is to be re-used for different tasks.



**Strengths And Weaknesses:**

Strengths:
  * The paper is extremely clearly written, it was very enjoyable to read!
  * The method is well-grounded in prior work and soundly derived theoretically
  * The experiments are well-chosen to demonstrate the functioning of the method and presented very clearly
  * In my opinion, this work is very relevant to the NeurIPS community, providing solid theoretical groundwork for model-based reinforcement learning approaches

Weaknesses:
I have some **minor** comments on the presentation of the experiments:
  * In Fig. 4b there seems to be a mismatch between the caption and the legend. Does "RL on real" stand for SAC?
  * In Fig. 6a it is not clear to me what Q-values are shown exactly

---

> ### Author Response · Authors · 2022-08-02
> **Response to HuRX**
>
> We thank the reviewer for their time reviewing our paper, and for the suggestions for improvement. We have run an additional experiment to study the reviewer's question about dynamics transfer and have revised the paper to incorporate the reviewer's suggestions. Below, we answer the reviewer's questions.
>
> > In Fig. 4b there seems to be a mismatch between the caption and the legend. Does "RL on real" stand for SAC?
>
> Yes. We have updated the legend.
>
> > In Fig. 6a it is not clear to me what Q-values are shown exactly
>
> The Q-values correspond to the discounted sum of returns under the current policy, $E_\pi\left[ \sum_{t=0}^\infty \gamma^t r(s_t, a_t) \mid s_0 = s, a_0 = a \right]$.
>
> > Would it be possible to combat the exploration problem by only modifying the reward with the classifier term for experience which comes from the model, and use the original reward for transitions observed on the real system?
>
> Yes, seems like a good potential solution. Currently, we're only training on simulated experience, but including experience from the original environment in that way might be a good way to mitigate the exploration problem.
>
> > if the authors could comment on if their proposed method also gives advantages in these situations where the model is to be re-used for different tasks.
>
> Yes, the model can be re-used, and we study this capability in a new experiment (Appendix Figure 13). Because our model is biased towards regions with high rewards, we expect that it will learn faster than an unbiased model on similar tasks (because the model optimism for one task will facilitate useful exploration for the new task) but will be slower when learning dissimilar tasks. We test this hypothesis by taking the dynamics model learned by MnM testing whether this dynamics model allows Q-learning to solve new tasks faster or slower, as compared with using the true environment dynamics. Again to avoid introducing approximation error, we used the stochastic gridworld (Fig. 2a). The results, shown in new Figure 13, show that the model bias does not hurt when solving easy or dissimilar tasks, but can accelerate learning when solving tasks that are both challenging and similar.

---

> > ### Comment · Reviewer_HuRX · 2022-08-07
> > **Response**
> >
> > Dear authors,
> >
> > thank you for the effort invested in clarifying my questions. I'll keep my initial score.

---

> > > ### Author Response · Authors · 2022-08-07
> > > **Thanks!**
> > >
> > > Thanks for the update! Please let us know if there are any additional questions or concerns!

---

### Meta-Review · Area_Chair_bmNH · 2022-08-27

**Recommendation:** Accept
**Confidence:** Certain

**Metareview:**

The paper addresses an important problem in model-based RL: the objective mis-match problem, in which the objective being optimized is different from the actual objective needed. The paper proposes a lower bound function on the expected return. The training resembles a GAN approach. The paper also presents numerical study to justify the method.

The reviewers unanimously agree that the paper could be accepted to NeurIPS. The reviewers appreciate the good writing of the paper and also recognize the technical novelty. However a major concern remains about the experiments. For instance, the reviewer believes there is a gap between the theory presented in the paper and the empirical experiments. It would be good to investigate deeper why the classifier term hurts exploration.

**Award:**

No

---

### Decision · Program_Chairs · 2022-09-14

Accept